# Composition of cetacean communities worldwide shapes their contribution to ocean nutrient cycling

Lola Gilbert [1,2], Tiphaine Jeanniard-du-Dot[1], Matthieu Authier[2], Tiphaine Chouvelon [2,3] & Jérôme Spitz [1,2]

Defecation by large whales is known to fertilise oceans with nutrients, stimulating phytoplankton and ecosystem productivity. However, our current understanding of these processes is limited to a few species, nutrients and ecosystems. Here, we investigate the role of cetacean communities in the worldwide biological cycling of two major nutrients and six trace nutrients. We show that cetaceans release more nutrients in mesotrophic to eutrophic temperate waters than in oligotrophic tropical waters, mirroring patterns of ecosystem productivity. The released nutrient cocktails also vary geographically, driven by the composition of cetacean communities. The roles of small cetaceans, deep diving cetaceans and baleen whales differ quantitatively and functionally, with contributions of small cetaceans and deep divers exceeding those of large whales in some areas. The functional diversity of cetacean communities expands beyond their role as top predators to include their role as active nutrient vectors, which might be equally important to local ecosystem dynamics.

The importance of cetaceans in marine ecosystem functioning has long been limited to their role as top predators and their top-down impact at lower trophic levels through prey consumption[1]. However, a growing body of research suggests that their importance in ecosystem dynamics extends to their role in fine-scale nutrient cycling processes[2–6]. The cycling of essential nutrients, such as nitrogen (N) or iron (Fe), is a key dynamic component of ecosystem functioning, from their initial uptake by primary producers to their transfer up the food web and their re-mineralisation. As air-breathing mammals, cetaceans urinate and defecate at the surface when they breathe or rest. This triggers patchy, transient nutrient enrichment events in the euphotic zone[7,8], which could locally stimulate marine productivity.

The euphotic zone tends to be naturally depleted in nutrients essential to phytoplankton growth, at the base of light-based trophic webs, resulting in conditions of suboptimal productivity in most of the world's oceans[9,10]. Productivity limitation is rarely due to the paucity of only one nutrient[11,12]. Trace nutrients (i.e. with low concentrations in

the environment), such as Fe, copper (Cu) or manganese (Mg), are essential cofactors of metalloenzyms involved in photosynthesis and respiration, or structural elements in proteins[12–14]. As such, they are as important in ecosystem functioning as major nutrients (N, phosphorous (P)), even if in substantially lower concentrations[10]. Oceans also consist of heterogeneous habitats with different intrinsic nutrient characteristics and associated nutrient limitation conditions[10]. At large spatial scales, primary productivity per unit area is two to ten times greater at high latitudes than at low latitudes[15]. Most low latitude systems are characterised by an intense water column stratification due to the elevated average temperatures[16], and phytoplankton biomass is primarily limited by nitrogen availability[10]. In contrast, higher latitudes are characterised by strong seasonal changes in their dynamics, and trace nutrient paucity is the primary factor limiting phytoplankton biomass[10]. At smaller spatial scales, neritic waters (i.e. coastal waters from the continental shelf) fed by terrestrial inputs tend to be eutrophic and have higher productivity per unit area than oceanic

[1]Centre for Biological Studies of Chizé, UMR 7372 La Rochelle University - CNRS, La Rochelle, France. [2]Pelagis Observatory, UAR 3462 La Rochelle University - CNRS, La Rochelle, France. [3]Ifremer, Chemical Contamination of Marine Ecosystems Unit, Nantes, France. ✉ e-mail: jspitz@univ-lr.fr

waters, which tend to be oligotrophic in the absence of major physical processes of nutrient enrichment (e.g. upwelling)[10,15]. Phytoplankton productivity is also the first biological process involved in the ocean carbon pump: carbon dioxide dissolved in surface waters is partly transferred to organic matter through photosynthesis, and partly sequestered in sediments via sinking particles and carcasses, passive advection and vertical migration of animals[17]. A thorough assessment of nutrient cycling in the euphotic zone is therefore essential to understanding the mechanisms regulating trophic web productivity and atmospheric carbon sequestration in the world's oceans, including nutrient cycling mediated by animals.

The ecological importance of animal-driven nutrient biological cycling is increasingly recognised and has been demonstrated in many ecosystems (e.g.[18–21]). Cetaceans are singular nutrient vectors in the oceans as (i) their waste products are greatly concentrated in nutrients compared to surface waters[7,8,22,23], (ii) they are highly mobile, and can transfer nutrients against physical forces and between habitats of different nutrient regimes[24,25], (iii) deep diving species can transfer nutrients from the ocean depths, where they feed, to the surface, mediating a nutrient pump known as the "whale pump"[7,26], (iv) some species can form large aggregations and create "hotspots" and "hot moments" of nutrient biological cycling[24,27], and, finally, (v) they are tied to the euphotic zone for breathing, where they release their wastes. On a large spatiotemporal scale, the gross nutrient enrichment caused by cetacean waste release is likely minor compared to that caused by physical processes (e.g. upwelling, weathering of shelf sediments)[28], or to the biological cycling by microfauna (microbial community, microzooplankton)[29]. However, it could be important in supporting certain ecosystem processes locally[24], disproportionately so in some contexts[30], although it is particularly challenging to investigate in the ocean realm.

In controlled laboratory conditions, nutrients leaching from cetacean wastes stimulate phytoplankton growth and productivity[8,31]. It is thus likely that nutrients released by cetacean communities in natura act as fertilizers and are - at least partially - being re-injected in marine ecosystems via phytoplankton uptake. Nutrients that are not directly taken up by primary producers might also be partially retained in the euphotic zone, and influence ecosystem productivity through indirect pathways. The microbial community (heterotrophic bacteria and viruses) could uptake some nutrients from cetacean wastes[32,33], and detritivore species of zooplankton could feed directly on faecal particles[7]. Microbial communities facilitate the rapid recycling of nutrients in the euphotic zone (microbial loop), and their dynamics are closely linked to those of primary producers[34]. Zooplankton also play an important role in nutrient biological cycling[35], and could facilitate the recycling of cetacean-released nutrients into organic matter, which in turn might affect the productivity and structure of communities at higher trophic levels[7]. Thus, the importance of cetacean-released nutrients in ecosystem functioning might not be limited to the fraction taken up directly by primary producers.

Estimating the total amount of nutrients that cetaceans release is, therefore, a good starting point for investigating their role in nutrient biological cycling, and has been a commonly used approach to date. However, the role of cetaceans in nutrient cycling in ocean surface waters and the importance of these processes in ecosystem functioning are still poorly understood on a worldwide basis. Previous studies were generally limited to single species - mostly large baleen whales, single location – mostly the Southern Ocean, and one nutrient - mainly iron or nitrogen[4,7,8,26]. Cetacean communities worldwide include species ranging from small porpoises to large rorquals. Cetacean species can be classically divided into three main guilds: small cetaceans (porpoise and dolphin species less than 4 m long and feeding in the epipelagic zone); deep divers (toothed cetaceans feeding in the meso- and bathypelagic zones, such as sperm whales, beaked whales, pilot whales and Risso's dolphin); and baleen whales (cetacean species in

which baleen plates take the place of true teeth and feeding in the epipelagic zone mainly on krill or small preys). These species have different metabolisms, behaviours, foraging ecologies and population sizes, and the composition of cetacean communities is likely to be different in contrasted environments. Hence, the diversity of the cetacean community could shape the spatial and temporal variability in the amount and quality of nutrients delivered by cetaceans to surface waters. Given the central role of phytoplankton in the productivity, regulation and resilience of marine ecosystems, the narrow understanding of the cetacean contribution to ocean nutrient cycling needs to be widened to a comprehensive view at broader taxonomic, spatial and nutrient-type scales.

This study provides a quantitative estimation of nutrients released by cetaceans at the community level in fourteen contrasted areas around the globe, for two major nutrients (N, P) and six trace nutrients (Fe, Cu, Mn, selenium (Se), zinc (Zn) and cobalt (Co)), and for a total of thirty-eight cetacean species. Our approach is based on a bioenergetic consumption-egestion/excretion model using prey nutrient concentrations, cetacean diets, metabolic parameters and cetacean abundance estimates from wide-scale multispecies surveys. We estimate both the quantities of nutrients released in cetacean wastes and the relative composition of these wastes. We show (i) how the cetacean community's contribution to nutrient cycling varies geographically from subarctic to tropical regions, and (ii) how the species composition of cetacean communities can shape these processes. We discuss the potential importance of cetacean nutrient release in ecosystem functioning in light of the functional characteristics of ecosystems and cetacean species. Taken together, the results of this work suggest that small cetaceans, deep-divers and baleen whales play different roles in nutrient biological cycling at both global and local scales and that diversity in the cetacean community may be important locally in shaping patterns of productivity and diversity in their ecosystems.

## Results

Model estimates (per area, habitat, taxa and taxa within habitats, waste relative composition) and Sobol sensitivity indices are provided in Supplementary Data 1, and results of statistical tests for differences (between areas, habitats and taxa) are available in Supplementary Data 2.

### Cetaceans release greater quantities of nutrients in temperate areas

The contribution of cetacean populations to nutrient cycling (annual quantity of nutrient released per surface unit) varies geographically from 2.1 kg.km$^{-2}$.yr$^{-1}$ (95% confidence interval - hereafter [CI 95%], i.e. [1.4; 3.1]) to 152.2 [96.9; 234.1] kg.km$^{-2}$.yr$^{-1}$ for the major nutrient N, and from 2.6 [1.2; 5.0] mg.km$^{-2}$.yr$^{-1}$ to 162.5 [80.8; 292.0] mg.km$^{-2}$.yr$^{-1}$ for the trace nutrient Co (Fig. 1). French Polynesia in the Pacific Ocean has the lowest release values for all nutrients, while the central North Atlantic Ocean shows the highest values, up to 77 times higher than French Polynesia's, for Fe release (Fig. 1). All tropical or sub-tropical areas (Gulf of Mexico, Hawaii, French Polynesia, New Caledonia, Wallis & Futuna, French Antilles and Guyana) show relatively low nutrient inputs from cetacean communities (2 to 10 times greater than French Polynesia, the baseline – poorest - area). In contrast, temperate and sub-Arctic areas (Gulf of Alaska, central North and Northeast Atlantic oceans) show high nutrient inputs from cetaceans (29 to 77 times greater than our baseline area). The Southwest Indian Ocean shows a relatively high nutrient release from cetaceans compared to other low-latitude areas (8 to 10 times greater than in French Polynesia). Among temperate areas, cetacean's nutrient release in the Mediterranean Sea is the lowest with fold-change ratios from 9 to 17, even when compared to areas at similar latitudes (California current and Northwest Atlantic, fold-change ratios from 8 to 45, Fig. 1).

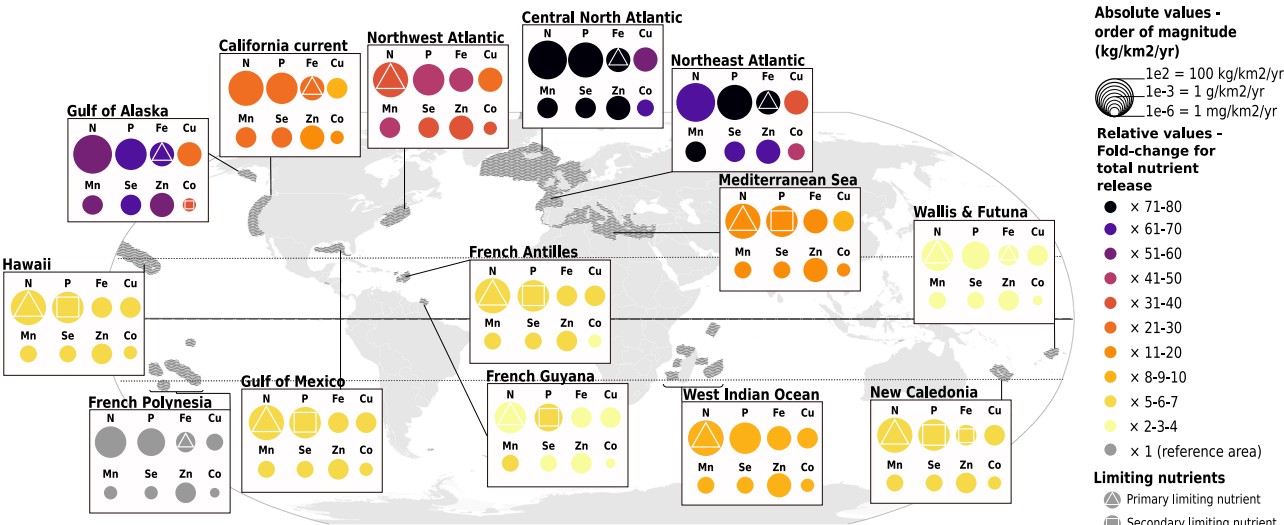

**Fig. 1 | Nutrient loads released by cetacean communities in 14 contrasted areas show quantitative and qualitative variations around the globe.** Results are from the bioenergetic model supplemented with an original dataset of abundance estimates, diet composition, prey composition and metabolic data. The model was set up with Monte-Carlo simulations combined with a bootstrap procedure with $n = 1^{e4}$. Dark grey shaded areas define locations surveyed for population abundance estimates used in the model. In each area (i.e. box), the sizes of the circles are proportional to the order of magnitude of mean absolute estimates in kg/yr/km$^2$; the colour gradient of the circles indicates values of the fold-change ratio of nutrient release compared to the area of reference (French Polynesia, where absolute values are the lowest), i.e. how much more nutrients are released in a given area compared to this baseline one. Shapes in the circles identify primary (triangles) and secondary (squares) limiting nutrients for primary producers in specific areas as taken from Moore et al. (2013)[10], Zhao & Quigg (2014)[73], Drupp et al. (2011)[74], Sonnekus et al. (2017)[75]. Vector map adapted from Felipe Menegaz/CC-BY SA 3.0/. Source data are provided as a Source Data file.

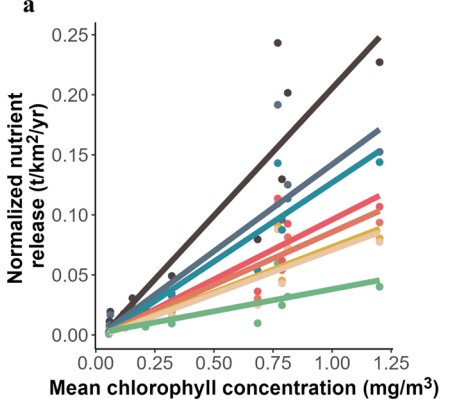

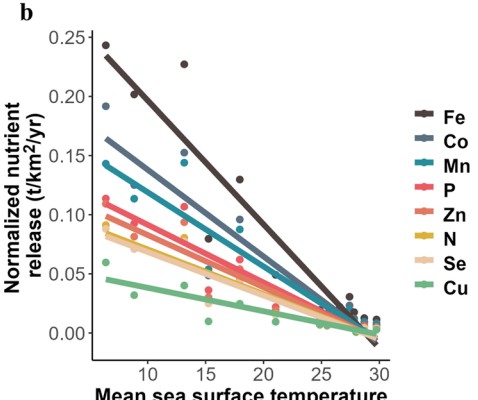

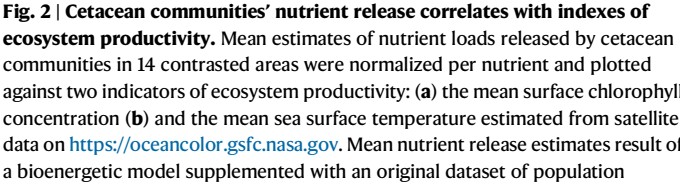

**Fig. 2 | Cetacean communities' nutrient release correlates with indexes of ecosystem productivity.** Mean estimates of nutrient loads released by cetacean communities in 14 contrasted areas were normalized per nutrient and plotted against two indicators of ecosystem productivity: (**a**) the mean surface chlorophyll concentration (**b**) and the mean sea surface temperature estimated from satellite data on https://oceancolor.gsfc.nasa.gov. Mean nutrient release estimates result of a bioenergetic model supplemented with an original dataset of population abundances, diet composition, prey composition and metabolic data and set up with Monte-Carlo simulations combined with a bootstrap procedure with $n = 1^{e4}$. French Guyana area was removed for the left plot as the chlorophyll concentration estimate was driven by the water turbidity due to the Amazon River plume. Linear models were run for each nutrient and each slope was statistically significant (see Table 1). Source data are provided as a Source Data file.

## The latitudinal nutrient release pattern correlates with productivity

Mean surface chlorophyll concentration and sea surface temperature (SST) are two correlated indicators of ecosystem productivity commonly used in habitat characterisation. The mean annual SST in the 14 studied areas ranged from 6.4 °C in the central North Atlantic to 29.8 °C in Wallis & Futuna, and the annual mean surface chlorophyll concentration from 0.06 mg.m$^3$ in French Polynesia to 1.20 mg.m$^3$ in the Northeast Atlantic Ocean (Fig. 2). Amounts of nutrients released by cetacean communities in the 14 studied areas are negatively correlated with the sea surface temperature ($R^2$ from 0.80 to 0.89 and $p$-value ≤ $1.5^{e-5}$), and positively correlated with mean surface chlorophyll concentration ($R^2$ from 0.65 to 0.89 and $p$-value ≤ $8.1^{e-4}$; Fig. 2, Table 1). Slopes of these relationships vary between nutrients, but they

are steeper with mean surface chlorophyll concentrations than with SST. For instance, the relationship between mean surface chlorophyll concentrations and nutrients is more than 5 times stronger for Fe than for Cu.

## Quantity of cetacean-released nutrients differ in neritic and oceanic waters but differences are not consistent between areas

Annual densities of cetacean-released nutrients are significantly higher in oceanic than in neritic waters in the Northeast Atlantic Ocean ($p ≤ 4.0^{e-4}$, $p$ referring to our calculated $p$-value as detailed in Methods), while the opposite is found for the Gulf of Alaska, the French Antilles and Guyana ($p ≤ 1.0^{e-4}$; Fig. 3). There is no significant difference in the quantities of all nutrients released between both habitats in the central North Atlantic Ocean and the Mediterranean Sea ($p ≥ 2.2^{e-1}$ and $7.5^{e-2}$,

**Table 1 | Linear regressions between absolute levels of nutrient release by cetacean communities and indicators of productivity (mean chlorophyll concentration Chlo and sea surface temperature SST)**

| Nutrient | covariate | slope | R2 | p-value |
|---|---|---|---|---|
| N | Chlo | 0.08 | 0.82 | 2.3e − 5 |
| P | Chlo | 0.10 | 0.85 | 7.0e − 6 |
| Fe | Chlo | 0.21 | 0.85 | 7.7e − 6 |
| Cu | Chlo | 0.04 | 0.65 | 8.1e − 4 |
| Mn | Chlo | 0.13 | 0.89 | 1.1e − 6 |
| Se | Chlo | 0.07 | 0.81 | 2.4e − 5 |
| Zn | Chlo | 0.09 | 0.81 | 2.7e − 5 |
| Co | Chlo | 0.14 | 0.78 | 5.4e − 5 |
| N | SST | −0.004 | 0.88 | 8.2e − 7 |
| P | SST | −0.005 | 0.88 | 7.4e − 7 |
| Fe | SST | −0.010 | 0.89 | 4.6e − 7 |
| Cu | SST | −0.002 | 0.80 | 1.5e − 5 |
| Mn | SST | −0.006 | 0.87 | 1.0e − 6 |
| Se | SST | −0.004 | 0.88 | 6.1e − 7 |
| Zn | SST | −0.004 | 0.88 | 8.0e − 7 |
| Co | SST | −0.007 | 0.86 | 1.5e − 6 |

*P*-values are provided in the summary of each linear regression output. Guyana was considered an outlier for its mean chlorophyll concentration due to turbidity caused by the Amazon River plume and was therefore removed from the data for the modelling analysis with the mean chlorophyll concentration as a covariate.

respectively). In the Northwest Atlantic Ocean area, Cu release is significantly greater in oceanic than in neritic waters ($p = 1.0^{e−3}$), while the difference is not significant for other nutrients ($p \geq 7.4^{e−2}$). In the Southwest Indian Ocean, nutrient release in oceanic waters is significantly greater than in neritic waters for all nutrients ($p \leq 4.5^{e−1}$) except P ($p = 1.1^{e−1}$), Fe ($p = 8.2^{e−2}$) and Mn ($p = 1.8^{e−1}$).

## Relative compositions of cetacean-released nutrient cocktails differ between ecosystems

In tropical and sub-tropical areas, relative proportions of nutrients within the same area are of similar range (differences in fold-change ratios are between 1 and 3, see homogeneous colours in Fig. 1). However, in temperate northern areas, nutrients can be over- or under-represented in the total load compared to the relative composition of the nutrient cocktail released in tropical areas. For example, in the Northeast Atlantic Ocean, cetaceans release 35 times more P and Fe than Cu (fold-change ratios of 72 and 37, respectively, Fig. 1). In the central North, Northeast and Northwest Atlantic oceans, the Gulf of Alaska and the California current, cetacean wastes are depleted in Cu and Co (lowest fold-change ratios in each area, between 9 and 54), while being enriched in Fe, P and Mn (highest fold-change ratios in each area, between 28 and 77) compared to tropical areas. Within a specific area, the relative composition of the nutrient cocktail released can differ between oceanic and neritic habitats (Fig. 3), e.g. in the Northwest Atlantic Ocean the difference between standardized release levels is around 0.5 for Cu while it is around 0.1 for Mn. Cetacean wastes are enriched in Cu in areas where cetaceans release more nutrients in oceanic habitats (highest difference compared to other nutrients in the Northeast & Northwest Atlantic oceans, and the Southwest Indian Ocean, Fig. 3).

## Relative contributions of baleen whales, deep divers and small cetaceans populations to whole community-released nutrients differ quantitatively and qualitatively

Multivariate analyses on the relative composition of wastes showed segregation of nutrient release profiles among the three cetacean groups, i.e. small cetaceans, deep divers and baleen whales, mainly

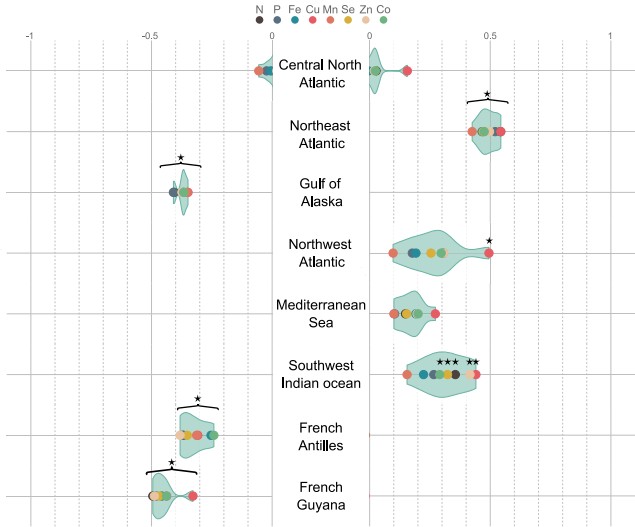

**Fig. 3 | Cetaceans do not release equivalent amounts of nutrients in different habitats, depending on areas.** Differences between mean levels of nutrients released by cetacean communities in neritic and oceanic habitats, with levels of nutrient release per unit area and per year normalized per nutrient and per area. When values are negative (left panel), nutrient release is greater in neritic than oceanic habitats, and when values are positive (right panel), nutrient release is greater in oceanic than in neritic habitats. Habitat differences between nutrient release per unit area and per year estimates were assessed based on unilateral binary relations between estimates (see Methods), and significant differences between habitats are indicated with a black star (test results are provided in Supplementary Data 2). Mean nutrient release estimates result of a bioenergetic model supplemented with an original dataset of population abundances, diet composition, prey composition and metabolic data and set up with Monte-Carlo simulations combined with a bootstrap procedure with $n = 1^{e4}$. Green shades are violin plots, indicating the distribution of difference estimates. Source data are provided as a Source Data file.

driven by the differential composition in Cu, P and Mn (Fig. 4). Deep divers release waste matter significantly richer in Cu (9.5 [2.6; 18.5] mg.Cu.kg$^{−1}$ of food ingested) than small cetaceans (3.5 [0.6; 10.6] mg.Cu.kg$^{−1}$ of food ingested, $p = 2.2^{e−2}$) and baleen whales (2.9 [0.6; 7.0] mg.Cu.kg$^{−1}$ of food ingested, $p = 1.4^{e−2}$; Fig. 5). No other significant difference for nutrients considered separately was found despite slight variations (e.g. higher P and Mn content in matter released by small cetaceans, $p \geq 4.2^{e−1}$).

These qualitative differences at species levels combined with release rates and population abundances result in differences in the relative contribution of baleen whales, deep divers and small cetaceans to total nutrient loads released by communities in the different studied areas (Fig. 6). Baleen whales contribute over 90% of cetacean-released nutrients in the Gulf of Alaska, about half (46–55%) in the Northeast Atlantic Ocean and 37 to 65% in the central North Atlantic Ocean. Their contribution is significantly greater than that of small cetaceans and deep divers for all nutrients only in the Gulf of Alaska ($p \leq 3.0^{e−4}$, Supplementary Data 2), and than that of deep divers for N in the Northeast Atlantic Ocean ($p = 1.3^{e−2}$). In the California Current, Northwest Atlantic Ocean and Mediterranean Sea, baleen whales contribute to total nutrient release less (30–41%, 22–35% and 17–26%, respectively) than small cetaceans (51–64%, 35–50% and 50–74%, respectively), except for Cu in the California current (49% released by baleen vs 30% by small cetaceans) and in the Northwest Atlantic Ocean (22% released by baleen whales vs 19% by small cetaceans; Fig. 6). Differences between the two groups are significant only in the Mediterranean Sea for N, P, Fe, Mn and Zn ($p \leq 4.0^{e−4}$, Supplementary Data 2). Deep divers are the least contributing group in the Gulf of Alaska, California Current, Mediterranean Sea and Northeast Atlantic Ocean, but they contribute

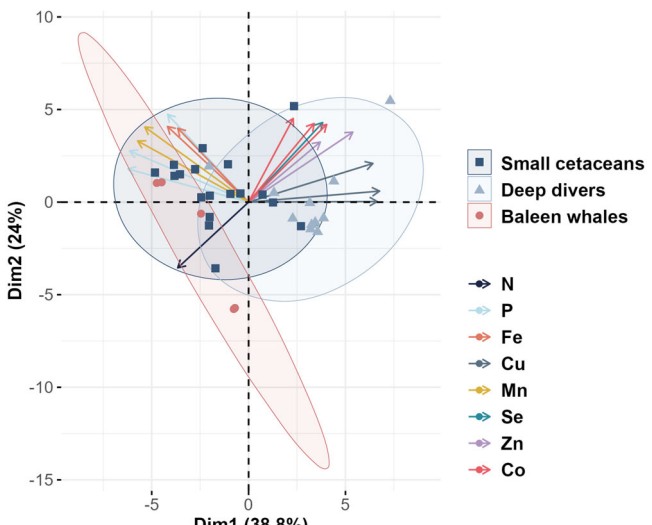

**Fig. 4 | Principal component analysis (PCA) reveals distinction between the relative nutrient composition of wastes released by small cetaceans, deep divers and baleen whales.** Individual nutrient released per kilogram of food ingested daily was estimated using a bioenergetic model supplemented with an original dataset of diet composition, prey composition and metabolic data and set up with Monte-Carlo simulations combined with a bootstrap procedure with $n = 1^{e4}$, normalized per nutrient and computed for 38 cetacean species belonging to small cetaceans (deep blue ellipse and square points), deep divers (light blue ellipse and triangle points) or baleen whales (red ellipse and circle points). Each point represents a species. The contribution of variables (lowest 2.5% quantile, mean and highest 97.5% quantile, for all nutrients) to the first two principal components are plotted as arrows on the biplot, colour indicates the nutrient. Only variables with cos2 > 0.5 were plotted. Source data are provided as a Source Data file.

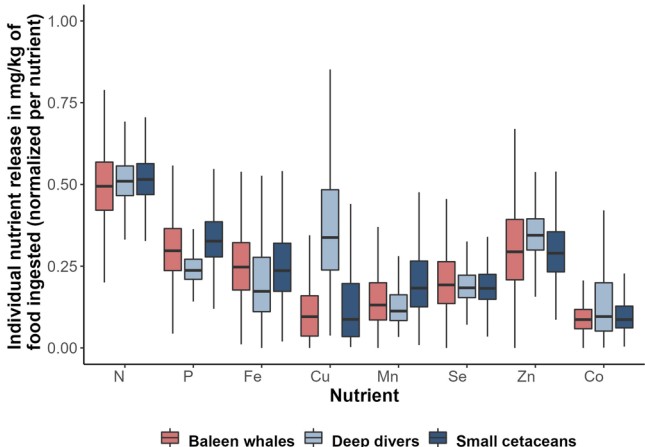

**Fig. 5 | Relative nutrient composition of wastes produced by small cetaceans, deep divers and baleen whales.** Individual nutrient released per kilogram of food ingested daily was normalized per nutrient and computed for 38 cetacean species, as estimated using a bioenergetic model supplemented with an original dataset of diet composition, prey composition and metabolic data and set up with Monte-Carlo simulations combined with a bootstrap procedure with $n = 1^{e4}$. For all nutrients except copper (Cu; with our calculated $p$-value $p = 2.2\ 1^{e\text{-}2}$ for comparison with small cetaceans) there is no significant difference between the relative composition of each taxon. Boxplots display the median with a solid black line in each box, lower and upper hinges correspond to the 25th and 75th percentile, respectively; upper and lower whiskers extend respectively from the hinges to the largest and lowest values no further than 1.5 times the inter-quartile range, and data beyond the end of whiskers are not plotted. Source data are provided as a Source Data file.

to 22–62% of the total loads in the central North and Northwest Atlantic oceans, where their contribution is not significantly different from that of baleen whales for all nutrients ($p \geq 9.2^{e-2}$) but P ($p = 4.3^{e-2}$) in the central North Atlantic Ocean (Fig. 6). Baleen whales are minor contributors to nutrient cycling in tropical Gulf of Mexico and Hawaii (0–4%, $p \leq 2.3^{e-3}$), where some species are present year-round and thus observed during abundance surveys (surveys are usually conducted in the summer season when migrating species are in their foraging grounds). Small cetaceans are the main contributors for all nutrients (62–74%, $p \geq 4.0^{e-2}$) but Cu (50%, $p \geq 6.2^{e-2}$), Se and Co (58–61%, $p \geq 5.1^{e-2}$ with baleen whales) in the Mediterranean Sea, and for all nutrients (74–85%, $p \leq 3.4^{e-2}$) but Cu and Co (42–68%, $p \geq 1.1^{e-1}$) in French Guyana. The contribution of deep divers is significantly the greatest in the cetacean community for all nutrients in New Caledonia (75–93%, $p \leq 2.3^{e-2}$), for N, P, Cu, Se, Zn in Hawaii (66–79%, $p \leq 3.4^{e-2}$), for N, Cu, Se and Zn in French Antilles (72–89%, $p \leq 4.8^{e-2}$), for Cu, Zn and Co in French Polynesia (70–87%, $p \leq 3.3^{e-2}$) and for Cu in the Southwest Indian Ocean (82%, $p = 1.0^{e-1}$). The most significant variations in relative contributions to the release of different nutrients are observed for Cu, for which the contribution of deep divers rises compared to other nutrients (e.g. from 31% for Fe to 62% for Cu in the central North Atlantic Ocean, or from 42% for Fe to 70% for Cu in Wallis & Futuna; Fig. 6). Deep divers contribute to more than 50% of Cu release by cetacean communities in 10 out of 14 areas, and to more than 70% in 7 of these areas. The relative contribution of small cetaceans is also the greatest for Mn and the lowest for Cu, in all areas.

Sobol sensitivity indices show that model outputs are most sensitive to population abundance (a fixed population abundance would reduce the output variance by up to 64%), followed by the metabolic multiplier $\beta$ (6–11% of output variance depending on nutrients), individual body mass (3–7% of output variance), and to a lesser extent by the mean energy content of the diet (2–5% of output variance; Fig. 7).

Diet mean nutrient content is more influential for trace nutrients than for major nutrients (e.g. fixing this parameter would reduce the output variance by 30% for Fe but only by 2% for N). The assimilation efficiency and nutrient release rates show little influence on the model output.

## Discussion

Cetacean communities release significantly more nutrients through their wastes in temperate latitudes than in tropical latitudes, mirroring patterns of ecosystem productivity (Figs. 1, 2). This illustrates a well-known bottom-up process: as predators depend on lower trophic levels, cetacean populations reflect the state of their environment[36]. Our results highlight how this bottom-up effect creates a 'nutrient virtuous cycle': the more productive the trophic base, the more nutrients are recycled through animal-mediated processes[37]. In addition, this study provides insights into the specific role of small cetaceans, deep divers and baleen whales in marine nutrient cycling, and shows that the role of small cetaceans and deep divers should not be overlooked. In most meso- to eutrophic areas, their relative contribution is up to that of baleen whales for most studied nutrients, even above in the Mediterranean Sea (Fig. 6). In tropical and subtropical waters, the productivity is too low to support populations of feeding baleen whales, leaving small cetaceans and deep divers as the only cetacean species involved in nutrient biological cycling in a large part of the world's oceans.

However, the varying amounts of nutrients returned to ecosystems by cetaceans cannot be considered proportional to their importance in ecosystem functioning, as the potential top-down effect of cetaceans on productivity does not necessarily scale with the bottom-up effect of primary productivity on cetacean populations. First, not all cetacean-released nutrients are equally likely to elicit a response from primary producers, depending on local demand. In most tropical and subtropical regions, where N and P are the primary limiting nutrients (Fig. 1), it is the N and P fractions of cetacean wastes that are likely to scale with ecosystem response. On the other hand, N

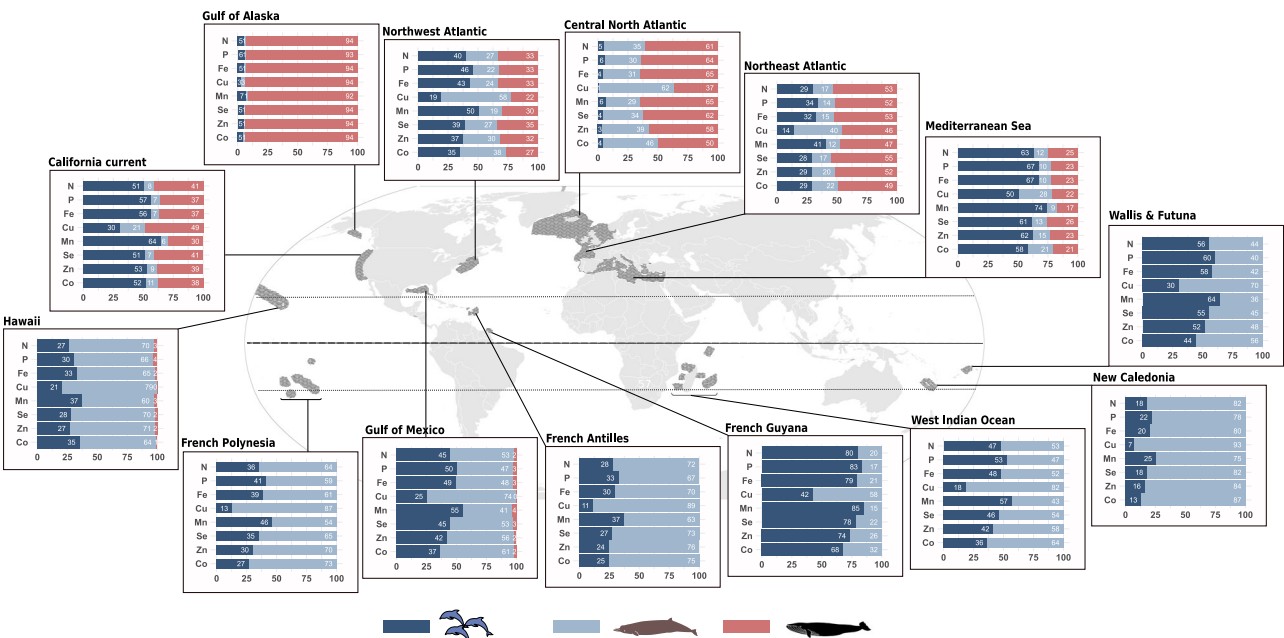

**Fig. 6 | Different cetacean taxa show different contributions to nutrient cycling worldwide.** Respective contribution (in %) of baleen whales (red), deep divers (light blue) and small cetaceans (deep blue) to loads of nutrients released by whole cetacean communities in 14 contrasted areas. Results are from the bioenergetic model supplemented with an original dataset of abundance estimates, diet composition, prey composition and metabolic data. The model was set up with Monte-Carlo simulations combined with a bootstrap procedure with $n = 1^{e4}$. Vector map adapted from Felipe Menegaz/CC-BY SA 3.0/. Source data are provided as a Source Data file.

and P released in cetacean wastes in High Nutrient Low Chlorophyll areas (HNLC, e.g. Southern Ocean or large regions of the Pacific Ocean) only add to their already high concentrations in these waters, whereas the released Fe could enhance the growth and productivity of primary producers, and thus be more likely to influence ecosystem functioning. It is interesting to note that the relationship between levels of cetacean-released nutrients and local productivity indicators is the strongest for Fe (Fig. 2, Table 1), identified as either a primary or secondary limiting nutrient in half of the fourteen areas included in the study, and in a large part of the world's oceans (Fig. 1). At the finer habitat scale, oceanic species of diatoms in temperate, meso- to eutrophic regions require higher Cu concentrations compared to their neritic counterparts, especially in iron-limited areas, due to the role of Cu in Fe acquisition[38,39]. Interestingly, nutrient loads released by cetaceans are richer in Cu in oceanic habitats than in neritic ones (Fig. 3). This is likely due to deep divers, largely associated to deep oceanic waters, that release waste products enriched in Cu (Figs. 4–6). Although not all cetacean-released nutrients are equally relevant to ecosystem functioning, the simultaneous release of nutrients within a highly concentrated cocktail could facilitate the supply of nutrient ratios optimal for primary producers[40] and have a synergistic effect by limiting the risk of co-limitation[41].

Furthermore, cetacean-mediated nutrient cycling is probably as important in ecosystem functioning as their relative contribution to shaping the nutrient background in their environment. Primary producers may rely more on animal-mediated nutrient cycling in oligotrophic ecosystems than in meso- to eutrophic ecosystems, which benefit from other sources of supply[24,40]. The contribution of cetacean communities to nutrient and ecosystem dynamics may be greater where nutrient sources are limited and water column stratification is intense, i.e. tropical systems and oceanic waters. This further highlights the potential importance of small cetaceans and deep divers in nutrient-biological cycling in the world's oceans. In temperate, meso- to eutrophic areas, the ecological relevance of cetacean-led nutrient cycling is likely more context-dependant[40] and relative to other

nutrient inputs. In the Northwest Atlantic Ocean, our estimate of cetacean N release (75 [52; 112] kg.N.km$^{-2}$.yr$^{-1}$; Supplementary Data 1, Table 2) is lower than that of Roman & McCarthy (2010)[7] for cetaceans in the Gulf of Maine (around 190 kg.N.km$^{-2}$.yr$^{-1}$; Table 2). These estimates are still of a similar order of magnitude than supplies from physical processes (e.g. river discharge and atmospheric deposition, Table 2). In the Mediterranean Sea, cetacean-released N (52,290 [35,250; 74,770] t.N.yr$^{-1}$) is equivalent to the natural N background of the Rhône or the Ebro river (52,339 t.yr$^{-1}$ and 51,018 t.N.yr$^{-1}$, respectively), while cetacean-released P (5650 [3740; 8320] t.P.yr$^{-1}$) is above natural weathering P input of the Rhône, Ebro, Pô and Evros rivers combined[42] (2782 t.P.yr$^{-1}$; Table 2). In the California current area, N released by cetaceans (49,270 [30,880; 75,820] t.N.yr$^{-1}$) is 5 times greater than the input from local rivers (10,000 t.N.yr$^{-1}$) and 6% of the input from seasonal upwelling (800,000 t.N.yr$^{-1}$)[43]. This demonstrates the potential ecosystemic value of cetacean-mediated nutrient cycling, although comparison is possible only for a few areas and nutrients.

Characteristics of cetacean-released nutrient cocktails – such as absolute quantities and relative nutrient composition, both quantified here, but also nutrient turnover rates in ecosystems, nutrient allochthonous/autochthonous status, and nutrient biochemical properties (sinking rate, chemical speciation) – also shapes their importance in ecosystem functioning. The scale at which individual small cetaceans, deep divers and baleen whales release nutrients is as different as their respective body mass ranges and metabolism, which suggests differences in the magnitude of the ecosystem response to defecation by the three taxa. Small cetaceans and deep divers, however, have higher consumption rates per unit body mass (Supplementary Data 3) and therefore release more nutrients per unit of cetacean biomass than baleen whales. This means that for an equivalent total biomass, a population of small cetaceans would release more nutrients than a population of whales. In addition, our results show differences in the relative composition of waste products released by small cetaceans, deep divers and baleen whales. Primary producer communities respond differently when fertilised with nutrient cocktails of different

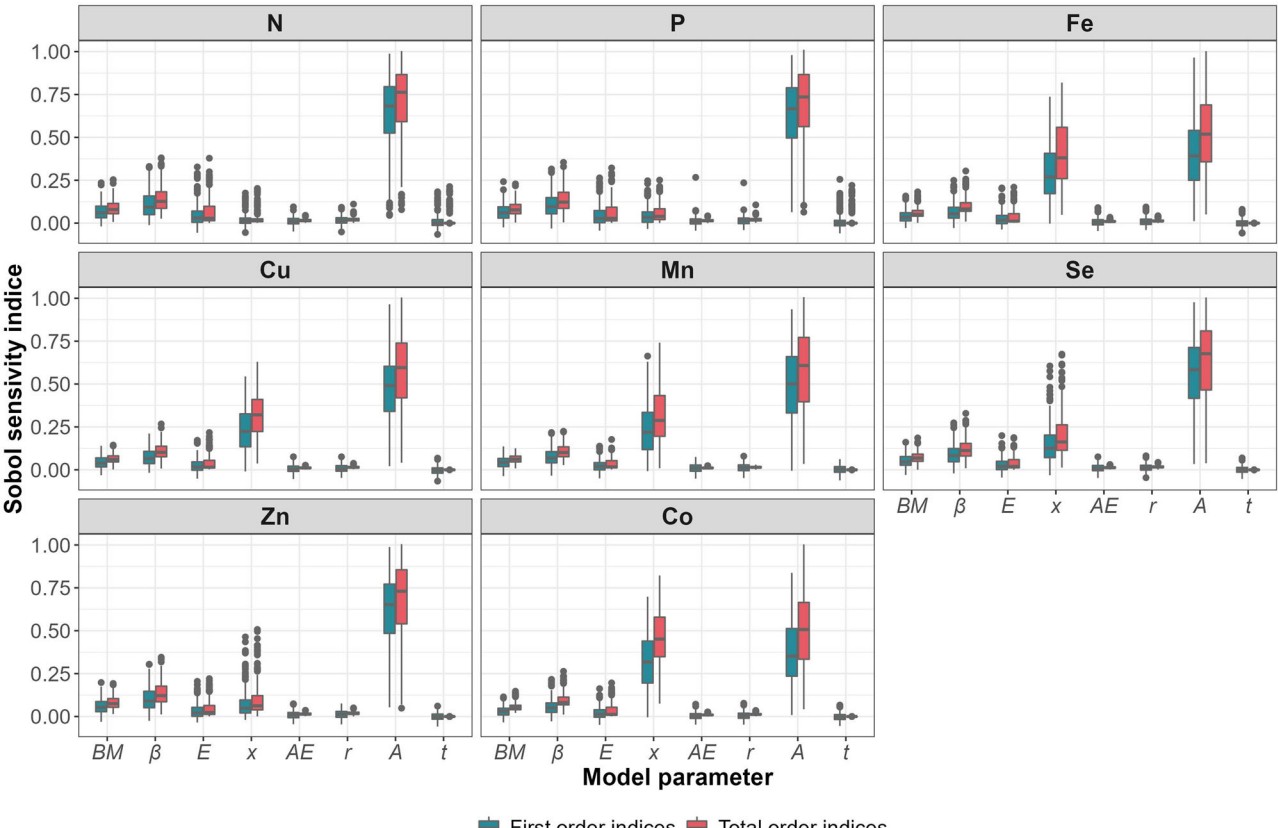

**Fig. 7 | Sensitivity of the bioenergetic model output to uncertainty in model inputs.** Ranges of estimates of Sobol indices (first-order, in blue, and second-order, accounting for interactions between parameters, in red) calculated for each species of cetacean included in the model, in each habitat of each area. BM is the body mass, β is as species-specific metabolic multiplier, E is the mean energy content in the diet, x is the mean nutrient content in the diet, AE is the mean assimilation of energy rate, r is the nutrient release rate, A the population abundance and t the number of days of presence. The bioenergetic model was set up with Monte-Carlo simulations combined with a bootstrap procedure with $n = 1^{e4}$. Boxplots display the median with a solid black line in each box, lower and upper hinges correspond to the 25th and 75th percentile, respectively; upper and lower whiskers extend respectively from the hinges to the largest and lowest values no further than 1.5 times the inter-quartile range, and data beyond the end of whiskers are not plotted. Source data are provided as a Source Data file.

composition[14,41,44]. Thus, ecosystem response to defecation by the three taxa could vary both quantitatively and qualitatively, given similar conditions in the recipient ecosystem. The composition of the potential fertiliser released reflects individuals' diet and the nutrient content of their prey (i.e. nutrient concentrations and bio-availability)[45], which in turn depends on prey taxa and habitat[46,47]. For example, Cu is a constituent of hemocyanin, the respiratory pigment of cephalopods and crustaceans. The high consumption of cephalopods by deep diving species (e.g. pilot whales *Globicephala*) results in higher levels of Cu in their diet[47] - and thus their waste products (Supplementary Data 1 & 3, Figs. 4, 5), compared to other species. Bony fish have high P contents, which is reflected in higher P levels estimated in the waste of small cetaceans. This illustrates the potential consequences of prey switching (in response to changes in prey quality and/or availability) on the nutrient-biological cycling mediated by cetaceans in marine ecosystems. The fact that model outputs are more sensitive to diet nutrient concentration for trace nutrients than for major nutrients is likely due to their higher variability in the composition of functional prey groups[46]. Variations in the relative composition of released nutrient loads between taxa, areas and habitats could be underestimated here, due to extrapolations made to apply the model to numerous species in numerous areas. Particularly, we assumed a fixed diet per species, and the compositional dataset for prey compiled samples from a unique area (Northeast Atlantic). While we limited the extent of these biases using bootstrapping and functional prey groups to describe diets (see Methods), these parameters

should be more precisely and locally set for future replication of our approach at finer spatial scales. This could be especially informative in revealing fine-scale patterns of nutrient deposition.

Nutrients released by each taxon also differ in their turnover rates, i.e. the time between the assimilation of nutrients in organic matter and their return to ecosystems in a bioavailable form. Even at low concentrations, a rapid turnover can support a greater fraction of primary producers than a slower one[48], and thus have a greater impact on the trophic web structure and productivity. Most species of baleen whales feed entirely or partially on zooplankton (Supplementary Data 4), i.e. either direct grazers of primary producers or predators of grazers. Thus, the turnover of nutrients released by baleen whales feeding on zooplankton is short compared to that of small cetaceans or deep divers feeding on fish and/or squid species, at much higher trophic levels. In regions such as the Gulf of Alaska or the Northeast and central North Atlantic oceans, where baleen whales are major contributors to the nutrient load released by the cetacean community (Fig. 6), the rapid turnover of the nutrients they release could further increase their importance in maintaining primary productivity levels.

Moreover, the origin of the released nutrients, i.e. inside the euphotic zone (true recycling) or outside of the euphotic zone (new supply), will affect their potential contribution to ecosystem productivity[49]. Recycled nutrients might help to maintain primary productivity levels, but new nutrients might stimulate new production. Baleen whales and small cetaceans mostly feed in the euphotic zone[50] and mediate the recycling of autochthonous nutrients within it. Deep

**Table 2 | Nitrogen and phosphorous released by cetacean communities in different study areas (in bold) compared to estimates of nitrogen and phosphorous inputs from other processes and sources**

| Area | River | Nutrient | Process | Amount (tonne/yr) | Reference |
|---|---|---|---|---|---|
| Mediterranean Sea | Rhône | N | Natural background | 52,339 | 42 |
| Mediterranean Sea | Rhône | N | Erosion | 24,427 | 42 |
| Mediterranean Sea | Ebro | N | Natural background | 51,018 | 42 |
| Mediterranean Sea | Ebro | N | Erosion | 14,607 | 42 |
| Mediterranean Sea | Pô | N | Natural background | 38,978 | 42 |
| Mediterranean Sea | Pô | N | Erosion | 27,755 | 42 |
| Mediterranean Sea | Evros-Maritsa | N | Natural background | 22,987 | 42 |
| Mediterranean Sea | Evros-Maritsa | N | Erosion | 6676 | 42 |
| Mediterranean Sea | | **N** | **Cetacean waste products release** | **52,290 [35,250; 74,770]** | **This study** |
| Mediterranean Sea | Rhône | P | Natural weathering | 883 | 42 |
| Mediterranean Sea | Rhône | P | Erosion | 9812 | 42 |
| Mediterranean Sea | Ebro | P | Natural weathering | 932 | 42 |
| Mediterranean Sea | Ebro | P | Erosion | 6275 | 42 |
| Mediterranean Sea | Pô | P | Natural weathering | 635 | 42 |
| Mediterranean Sea | Pô | P | Erosion | 8262 | 42 |
| Mediterranean Sea | Evros-Maritsa | P | Natural weathering | 332 | 42 |
| Mediterranean Sea | Evros-Maritsa | P | Erosion | 2961 | 42 |
| Mediterranean Sea | | **P** | **Cetacean waste products release** | **5650 [3740; 8320]** | **This study** |
| Gulf of Maine (103,000 km²) | All rivers | N | Total *N* | 11,200 | 7 |
| Gulf of Maine (103,000 km²) | | N | Atmosphere | 130,260 | 7 |
| Gulf of Maine (103,000 km²) | | **N** | **Cetacean waste products release** | **19,600** | 7 |
| Extended Gulf of Maine (451,985 km²) | | **N** | **Cetacean waste products release** | **34,400 [23,530; 50,600]** | **This study** |
| California Current | | N | Upwelling | 750,000 | 43 |
| California Current | All rivers | N | Total *N* | 10,000 | 43 |
| California Current | | **N** | **Cetacean and seals waste products release** | **49,270 [30,880; 75,820]** | **This study** |

divers, on the other hand, forage in deeper water layers, routinely below the euphotic zone (around 200 m depth) and as far down as the bathypelagic zone, at depths of up to 3000 m depth, depending on species. They operate a vertical nutrient transfer, bringing allochthonous nutrients back to the euphotic zone through their wastes[15]. In Hawaii, New Caledonia, the French Antilles or French Polynesia, 60 to 80% of the nutrients released by the cetacean community are newly introduced to the euphotic zone through this nutrient pump (Fig. 6). This highlights the importance of the nutrient supply provided by deep divers in these oligotrophic regions, where water stratification is intense and growth conditions for primary producers are especially limiting.

An issue that remains unresolved is whether wastes produced by the different cetacean species have the same biochemical properties. The chemical speciation of released nutrients and how they are retained in the euphotic zone, i.e. how much and for how long, are likely to be primary determinants of the ecological importance of the functional characteristics discussed above. Collecting waste from small cetaceans and deep divers is even more challenging than it is already for baleen whales, but experimental designs using products collected from stranded animals may help to reveal their distinctive properties. Such studies could be particularly instructive and would shed new light on current findings and discussions.

Finally, the scale of annual estimates over broad regions obscures the variations in spatial and temporal patterns of nutrient deposition. The actual relative contribution of small cetaceans, deep divers and baleen whales is unlikely to be constant throughout the year. Baleen whales migrate annually between oligotrophic breeding grounds, where they fast, and meso- to eutrophic temperate or polar regions, where they take advantage of the high productivity season to feed intensively[50]. In contrast, small cetaceans and deep divers feed at a fairly constant rate throughout the year and, although their distribution may vary slightly between seasons (e.g.[51]), they do not perform long-range migrations. Taking the central North Atlantic Ocean area as an example, the relative contribution of baleen whales to Fe release increases from 65% over a year to 86% over a four-month presence period. With the additional contributions of small cetaceans and deep divers over the same period, this means that 76% of the Fe released in a year is actually released in just four months, during the high productivity season. In subtropical and tropical areas, however, nutrient release rates and taxa relative contributions are likely to be fairly constant throughout the year, reflecting the stability of environmental conditions in these regions. The temporal component of these nutrient release patterns is thus important to bear in mind.

Similarly, cetaceans do not uniformly release nutrients across the large areas in which they live. Different species tend to have different habitat preferences and more or less patchy distributions, so they are likely to release their waste in areas with different intrinsic characteristics. Moreover, the size of their aggregations influences the intensity of the nutrient uptake and release events[24,25]. As such, their aggregation behaviours when they forage, travel, socialise or rest near the surface will determine whether they disperse or concentrate nutrients in their environment. For example, deep divers are thought to disperse when foraging at depth[52], but several species form aggregations near the surface, where they release their waste, presumably concentrating nutrients in their environment. These aggregations can range from a few individuals (e.g. beaked whales) to a dozen (e.g. sperm whales *Physeter macrocephalus* or Risso's dolphins *Grampus griseus*) or even a few hundred (e.g. pilot whales). In contrast, aggregations of baleen whales are more likely to occur when foraging (e.g. supergroups of fin whales *Balaenoptera physalus*[27]) than when resting, probably facilitating the dispersal of nutrients ingested in nutrient hotspots. Small

**Table 3 | Parameter settings of the bioenergetic model used to estimate nutrient release by cetacean waste products using Monte-Carlo simulations ($n = 1^{e4}$)**

| Parameter | Monte-Carlo simulation setting |
|---|---|
| Body mass BM | $\sim N(\overline{BM}, 0.2 * \overline{BM})$ |
| Beta β | $\sim N(\bar{\beta}, 0.2 * \bar{\beta}, a = 1, b = 5)$ |
| Assimilation efficiency AE | $\sim N(0.85, 0.05, a = 0.8, b = 0.95)$ |
| Nutrient release rate $r_n$ | $\sim U(0.2, 0.4)$ for N and P for migratory baleen whale species in their breeding grounds<br>0 for Fe, Cu, Mn, Se, Zn, Co for migratory baleen whale species in their breeding grounds<br>$\sim U(0.7, 0.9)$ in all other cases |
| Population abundance A | $\sim \log N\left( \log\left( \frac{\bar{A}}{\sqrt{1 + (C\overline{V}_A)^2}} \right), \sqrt{\log(1 + (C\overline{V}_A)^2)} \right)$ |
| Number of days of presence t | $\sim U(120, 240)$ for migratory baleen whale species in their feeding grounds<br>365 in all other cases |

Sources are provided in Supplementary Data 4.

cetaceans, on the other hand, are commonly observed in groups both when foraging and when resting or socialising near the surface. Although they are less likely to generate nutrient concentration or dispersal patterns, they still facilitate horizontal transfers. This can be particularly important for ecosystem functioning when the intrinsic nutrient characteristics of the donor and recipient ecosystems are significantly different[24], e.g. diurnal movements of spinner dolphins (*Stenella longirostris*) between offshore waters, where they feed, and lagoons, where they rest, in the Maldives and the Chagos archipelagoes[53].

To conclude, this study provides a broad, worldwide view of cetacean contribution to nutrient biological cycling. Cetacean-mediated nutrient cycling in ocean surfaces varies geographically, quantitatively and qualitatively, and at different spatial and temporal scales. Variations are largely driven by the specific composition of cetacean communities. The complexity of the processes involved renders the role and importance of cetacean-mediated nutrient biological cycling in ecosystem functioning difficult to decipher, but we identified characteristics of both cetacean species and local ecosystems that could matter. Differences in the characteristics of recipient ecosystems, cetacean wastes, and patterns of nutrient deposition accumulate over space and time. Together, they shape the role of small cetaceans, deep divers and baleen whales in ocean nutrient cycling, and determine the importance of these processes in ecosystem functioning. There is still a long way to go in quantifying cetacean contribution to the marine biological carbon pump, but it is difficult to fathom this ecosystem service being replaced by large-scale artificial fertilisation, once considered as a geoengineering solution to obtain carbon credits[54]. Furthermore, the different functional traits of small cetaceans, deep divers and baleen whales mean that the loss of a given population can hardly be substituted by an equivalent biomass of another taxon without altering the role of the community. Thus, the functional diversity of the cetacean community, largely known for their role as top predators, also applies to their role as active nutrient vectors and may be equally important for local ecosystem dynamics. In the current era of climate change, biodiversity loss, and trophic and habitat simplification, our results support the importance of maintaining and restoring healthy, diverse and abundant cetacean populations in the world's oceans.

## Methods

### Bioenergetic model
We used a bioenergetic model of individual consumption previously applied to marine mammal populations[55–57], and adapted it to estimate the nutrient consumption and egestion of populations[7]. The model estimates $Q_{tot,n}$, the yearly amount of nutrient $n$ (in t.$year^{-1}$) released by a given cetacean species in a specific area using Eq. (1) below, in which the BMR is the Basal Metabolic Rate of individuals

($BMR = \beta \times 293.1 \times BM^{3/4}$ in $kJ.day^{-1}$, with body mass BM in $kg$ and $\beta$ a species-specific metabolic multiplier accounting for additional cost of daily activities[57–59], E is the mean diet energy content (in $kJ.kg^{-1}$ fresh weight), AE the digestive assimilation efficiency, $x_n$ the average concentration of nutrient $n$ in the diet (in $mg.kg^{-1}$ fresh weight), $r_n$ the release rate of nutrient $n$, $t$ the number of days of presence in an area within the year and $A$ the population abundance:

$$Q_{tot,n} = \frac{BMR}{AE \times E} \times x_n \times r_n \times t \times A \times 1^{e6} \qquad (1)$$

The model calculations were carried out in fourteen areas where large-scale, multi-species cetacean population abundance estimates were available (Supplementary Data 4). Thirty-eight cetacean species were included.

### Model uncertainty
All parameters are associated with a certain degree of uncertainty and/or inherent natural variability. To account for it, we combined Monte-Carlo simulations and bootstrapping ($1^{e4}$ simulations and drawings, respectively) to simulate vectors of possible values for each parameter based on a given basal value and distribution (Table 3). Six model parameters (BM, β, AE, $r_n$, A, t) were sampled from parametric statistical distribution based on published information (Table 3). Two (E and $x_{Fe}$) were estimated based on the diet of each species and the composition of prey items: uncertainty was quantified using the bootstrap. We found high ranges of variations for some model parameters (Supplementary Data 3), which ultimately impacted uncertainty in the model output.

### Model parameters
The mean body mass $\overline{BM}$ used in the model resulted from body length to body mass regression equations[60]. For species not listed in Trites & Pauly (1998)[60], we either used other published reference or the mean body mass of a morphologically similar species (Supplementary Data 4). We considered a standard deviation of 20% of $\overline{BM}$ using a normal (Gaussian) distribution to account for intra-species variability and uncertainty (Table 3).

The metabolic index $\beta$ is a species-specific indicator of the "cost of living": it accounts for the cost of activities and metabolic efficiency[59]. Variability for $\beta$ was simulated using a truncated normal distribution (Table 3), with different base values depending on species. We used three base values $\bar{\beta}$: 2, 3, and 4. We associated a $\bar{\beta}$ of 2 to species with low cost of living (e.g. sperm whales), and a $\bar{\beta}$ of 4 to species with high cost of living (e.g. harbour porpoise). We defined three functional and ecological groups of cetaceans: baleen whales, deep divers and small cetaceans, partly guiding the choice of $\bar{\beta}$. We also considered that the

consumption of energy-rich food indicates high-energy needs and consequently a relatively high cost of living[59].

To account for the additional cost of lunge-feeding[5], $\bar{\beta}$ was increased by 0.5 for lunge-feeding baleen whale species. Individual daily ratios estimated by our model for these species (Supplementary Data 3) fall between previously published estimates[5]. For all species, we set the truncated normal distribution with minimum and maximum values of 1 and 5 to obtain physiologically plausible ranges (Table 3).

The mean assimilation efficiency of energy $AE$ is typically ~80% in cetacean bioenergetic models[7,56,57]. Yet, experimental studies gave values in the range of 73–93% for cetaceans[61,62]. To lean on the conservative side, we chose a base value of 85% and set the truncated normal distribution with minimum and maximum of 80% and 95% to avoid non-physiologically plausible values (Table 3).

The release rates of nutrients $r_n$ were never measured on cetaceans. For captive pinnipeds, N release rate ranged between 84 and 89%[63,64]. Most studies focusing on mammal species suggest high release rates for micronutrients such as Fe, Cu or Mn[45,65,66]. Bioenergetic models estimating nutrient egestion for cetaceans used a release rate of 0.80 for N[7], 0.80 for Fe or 0.85 again for Fe but with a range of variation between 0.70 and 0.90[26,67]. Given the little empirical information available, we defined the release rate as following a Uniform distribution between 0.7 and 0.9, for both major elements (N, P) and trace nutrients (Fe, Cu, Mn, Se, Zn, Co; Table 3). As migratory baleen whales fast in their breeding grounds, we considered these species only urinate during their presence there. Excretion of trace nutrients in urine is negligible[45,66], so we set nutrient release for the migratory species in breeding grounds to zero for trace nutrients and to between 0.2 and 0.4 for N and P (Table 3).

For population abundance $A$, we selected only dedicated large-scale, multi-species surveys using distance sampling protocols and analysis. When the survey design included several spatial blocks, we used mean estimates and coefficients of variation (CV) of blocks to compute the overall mean and CV for each area (Supplementary Data 4). Distinct estimates between habitats were computed where both oceanic and neritic blocks were surveyed. We then used a log-normal distribution with the calculated parameters for Monte Carlo simulations (Table 3).

Blue (*Balaenoptera musculus*), fin (*Balaenoptera physalus*), Sei (*Balaenoptera borealis*), Bryde's (*Balaenoptera edeni*) and humpback (*Megaptera novaeangliae*) whales are known to migrate from their feeding grounds of temperate and subpolar areas to breeding grounds in tropical to sub-tropical areas[50]. Of the fourteen areas included in this study, six are identified as feeding grounds (Northeast and Northwest Atlantic, Central North Atlantic, California current and Gulf of Alaska), while six are known breeding grounds (Hawaii, French Polynesia, Southwest Indian Ocean, French Antilles, New Caledonia and Wallis & Futuna). To account for migration behaviour, we included a parameter $t$ set between 120 and 240 days (four to eight months) following a uniform distribution in feeding and breeding grounds (Table 3). For other areas and species, $t$ was set as a constant at 365 days.

A unique diet was defined per species with no difference between locations and no seasonal variation, using diet composition data from the published literature (119 references in total, Supplementary Data 4). While we are aware that a unique diet could lead to an underestimation of the spatial variability of the final output, we used nine functional prey groups to describe diets to reduce this bias. If populations of the same species in different locations are unlikely to feed on the same prey species, they are likely to have the same trophic ecology, and therefore target prey from the same functional groups. We preferred information related to the percentage of biomass ingested of prey species or group of prey species (%W) from stomach content analysis (64 references) rather than qualitative information from isotope analysis, list of prey or surface observations (55 references). When not available, qualitative data was used to describe an average diet. We then compiled energy and nutrient content analytical data of 154 prey samples from two studies[46,47]. We were limited to these data sets in our analysis, but we artificially extended them by using a kernel-based bootstrap procedure to simulate variation in the composition of each prey group and obtain vectors of energy content ($E_{pg}$) and nutrient concentration ($x_{n,pg}$) per prey group $pg$. A distribution for the composition of each prey group (for each nutrient, separately) is estimated from sample values from the compiled data sets and $1^{e4}$ values are then drawn randomly from this estimated distribution. For the prey group *Zooplankton*, there was only one sample which precluded the use of the bootstrap procedure. We instead conducted Monte-Carlo simulations ($n = 1^{e4}$) using a normal distribution with the value of the sample as a mean and a standard variation of 20% for energy and macronutrient content and 40% for micronutrient content. Then, we used these energy and nutrient content values ($E_{pg}$ and $x_{n,pg}$, respectively) and the percentage of prey groups in diets ($W_{pg}$) to determine the mean energy content ($E$, Eq. (2)) and the mean iron concentration ($x_n$, Eq. (3)):

$$E = \sum_{pg} W_{pg} \times E_{pg} \tag{2}$$

$$x_n = \sum_{pg} W_{pg} \times x_{n,pg} \tag{3}$$

Datasets of prey composition are available under the depository system PANGEA[68,69] (https://doi.pangaea.de/10.1594/PANGAEA.937345 and https://doi.pangaea.de/10.1594/PANGAEA.940861). Further details and references for the setting of parameters are provided in Supplementary Data 4.

## Differences between areas, habitats and cetacean groups

Dimensions of parameters and outputs ($n = 1^{e4}$) resulting from the Monte Carlo simulations and bootstrap precluded the use of standard statistical tests of significance when comparing outputs for two groups (either areas, habitats, or taxonomical groups). Instead, we assessed unilateral binary relations by calculating the percentage of values from one group superior to the other group. We considered the difference significant when this percentage was ≥95% or ≤5%. To describe results, we either directly mentioned this *p*-value if it was ≤5% or expressed (1 − *p*-value) when it was ≥95% and adapted the direction of the binary relation of interest, so the result of our test (called *p*) could be expressed as in classic statistical tests (i.e. significant difference when $p \leq 0.05$). For example, if a test investigating whether nutrient release in area A is greater than nutrient release in area B and the result resulted in a 0.99 value, we concluded that nutrient release in area 1 is significantly greater than nutrient release in area B with $p = 0.01$.

## Sensitivity analysis

We conducted a sensitivity analysis to determine the parameters' influence on the final output value and uncertainty[70]. We adopted a global approach so that the effect of one parameter could be estimated when all other inputs were varying. It enables the identification of interactions and does not require the model to be linear and additive[71]. We used the Sobol variance-based approach where the variance of the output can be decomposed into the contributions imputable to each input factor.

## Nutrient release and ecosystem productivity

Surface chlorophyll concentration and sea surface temperature (SST) were extracted for our study areas (year 2021, https://oceancolor.gsfc.nasa.gov). We computed their mean values over each of our 14 studied areas and tested their relationships with cetaceans-released nutrient levels using linear models, this for the 8 nutrients.

## Relative composition of cetacean waste products

We estimated the stoichiometry of the waste products regardless of the amounts released by each species by calculating how much of each nutrient was released per kilogram of food ingested, and then by normalizing values per nutrient across species. Minimum, mean, lower and higher quantiles and maximum were calculated for small cetaceans, deep divers and baleen whales. We investigated the cetacean taxonomic influence on the composition of the nutrient cocktail released using Principal Component Analysis on summary statistics (quantiles and mean) of the relative composition of each species.

## Reporting summary

Further information on research design is available in the Nature Portfolio Reporting Summary linked to this article.

## Data availability

No original data were used nor generated for this analysis which mobilised data from the literature for the setting of a majority of parameters in our bioenergetic model. Sources are provided in the manuscript and in Supplementary Data 4. The latter also includes means and coefficient of variations used to define population abundances and the diet description used for each species using functional groups of prey. The external datasets of prey composition are available at https://doi.pangaea.de/10.1594/PANGAEA.937345 and https://doi.pangaea.de/10.1594/PANGAEA.940861 (see Methods). Oceanographic data was downloaded at https://oceancolor.gsfc.nasa.gov/l3/order/ in March 2022. The download procedure was indicated in a ReadMe file in the corresponding data folder, on the Github repository attached to the study (https://github.com/Lola-san/Cetacean.excretion.global.git). Other data used as an input in our model are also available at this link. The data generated in this study are provided in Supplementary Data 1. Source data are provided with this paper.

## Code availability

The code to reproduce the full analysis, originally performed using R Statistical Software (v.4.1.2)[72], are available on Github (https://github.com/Lola-san/Cetacean.excretion.global.git).

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

## Acknowledgements
This work was supported by the French Ministry for Ecology ("Ministre de la Transition Ecologique et Solidaire/Direction de l'Eau et de la Biodiversité - MTES/DEB") and by the European project H2020 SUMMER "Sustainable Management of Mesopelagic Resources" (grant agreement ID: 817806). We are grateful to Nicolas Bousquet for his guidance in the design of a bootstrap procedure appropriate for our sensitivity analysis.

## Author contributions
L.G., T.J.D.D. and J.S. designed the study; L.G. performed the data analysis with contributions of J.S. and T.J.D.D. on parameter settings, of T.C. on the use of prey composition data in the modelling design and of M.A. on the statistical implementation of the analysis; L.G., T.J.D.D. and J.S. wrote the manuscript.

## Competing interests
The authors declare no competing interests.
