## [Peer Review File · Nature Communications]

REVIEWER COMMENTS

Reviewer #1 (Remarks to the Author):

This paper takes on a broad, difficult, and important task of attempting to quantify the role of cetaceans in global ocean nutrient cycling. The authors take on this ambitious task across all clades of cetaceans and in all ocean basins. There are a lot of strong ideas and analyses in the present draft, but what is lost with an analysis at this level is the granular detail that may 'make or break' these findings. Below I discuss several major concerns with the present paper. After that I touch on minor concerns that are easily remedied.

One major concern I have is the spatial scales employed in this study. Research has shown that when viewed at the scale of oceans, it is hard to make top predator nutrient recycling "matter" when compared to abiotic sources of nutrient addition and mixing (Maldonado et al. 2016). Instead, cetaceans and other top predators, may initiate or enhance spatially restricted hot spots or "hot moments" (Pearson et al. 2023). I think adding that level of granularity to this manuscript would be immensely helpful. For example, large groups of baleen whales in the modern Southern Ocean can number several hundred to 1000 individuals (Herr et al. 2022, Ryan et al. 2023); how much trace metals might they recycle on the scale of days and tens of km? How do those levels compare to background trace metal concentrations in that same region at that same time of year without whales present?

At the scale the data is currently presented (vast ocean regions) I am left thinking the correlation of whale nutrient release and remotely-sensed chlorophyll is likely more bottom up than top down. Without examining this relationship in more detail, both spatially and temporally, it is hard to assign any importance to cetaceans. Meaning, at large spatial scales in an eastern boundary upwelling system, for example, upwelling itself drives the majority of primary productivity, and whales (and other predators) show up 1-2 months after peak phytoplankton blooms to forage on low trophic level prey that was fed by the upwelling-fueled bloom (Croll et al. 2005).

Additionally, ecosystem context is not provided in the level of detail it needs to be; in several cases this leads you to compare apples to oranges. For example, in lines 97-114, you compare nutrient release rates between micronutrients (e.g., trace metals) and macronutrients (N and P). These are not equivalent. One gram of bioavailable iron addition to iron-limited waters can stimulate several kilograms of primary production, the same is not true of N or P in waters limited by those macronutrients. If you plan to make these comparisons, it needs to be done in relation to the amount of primary production the levels of whale-recycled nutrients can seed.

Similarly, all these comparisons need to be done relative to the nutrient that is limiting in the ecosystem. This needs to be done on an ecosystem-by-ecosystem basis. For example, the additional N and P that whales recycle in the Southern Ocean is effectively meaningless in HNLC waters, whereas the amount of trace metals, specifically iron, they recycle (in a trace metal-limited system) could be immensely important. This is why there has been so much work done on the latter (Nicol et al. 2010, Lavery et al. 2010, 2014, Ratnarajah et al. 2014, Savoca et al. 2021), but not the former.

Finally, some relevant cetacean biology and ecology is lacking. The differences in annual phenology of cetacean feeding is glossed over, but is immensely important for this paper. Baleen whales have a compressed but very intense feeding season at high latitudes. Toothed whales generally eat similar amounts throughout the year, and across the world's oceans at all latitudes. Most baleen whales migrate; many toothed whales do not have the same predictable, long migrations. Point being, if you're looking at the total amount of nutrient(s) recycled on an annual basis and conclude that it is similar for baleen and toothed whales, but do not explicitly consider that baleen whales do nearly all their feeding in a 4-month period (vs spread out evenly across the entire year for most toothed whales), this may lead to erroneous conclusions about the "value" of that potential fertilizer.

Minor issues:

Line 13: "eight nutrient cycling" doesn't make sense as written.

Lines 30-34 & 318-321: you mention whales as a natural climate solution (NCS) while only mentioning carbon capture and storage. NCS only happens when carbon is sequestered for hundreds or thousands of years. This is an especially important component of this whole discussion and one that is unknown for whales. This section needs to be written much more speculatively.

Lines 62 and 64: "at large scale" and "at small scales" should likely be rephrased to: "at large/small spatial scales"

Lines 182-185: Oddly worded, suggest rephrasing

Lines 205: But N and P are macronutrients.

Lines 287-293: This is an essential point and needs to be mentioned much earlier in the paper (the introduction) and reiterated here.

Line 325: delete one "to"

Line 409: provide scientific names when first introducing species.

Literature cited

Croll, D., B. Marinovic, S. Benson, F. Chavez, N. Black, R. Ternullo, and B. Tershy. 2005. From wind to whales: trophic links in a coastal upwelling system. *Marine Ecology Progress Series* 289:117–130.

Herr, H., S. Viquerat, F. Devas, A. Lees, L. Wells, B. Gregory, T. Giffords, D. Beecham, and B. Meyer. 2022. Return of large fin whale feeding aggregations to historical whaling grounds in the Southern Ocean. *Scientific Reports* 12:9458.

Lavery, T. J., B. Roudnew, P. Gill, J. Seymour, L. Seuront, G. Johnson, J. G. Mitchell, and V. Smetacek. 2010. Iron defecation by sperm whales stimulates carbon export in the Southern Ocean. *Proceedings. Biological sciences / The Royal Society* 277:3527–31.

Lavery, T. J., B. Roudnew, J. Seymour, J. G. Mitchell, V. Smetacek, and S. Nicol. 2014. Whales sustain fisheries: Blue whales stimulate primary production in the Southern Ocean. *Marine Mammal Science* 30:888–904.

Maldonado, M. T., S. Surma, and E. A. Pakhomov. 2016. Southern Ocean biological iron cycling in the pre-whaling and present ecosystems. *Philosophical Transactions of the Royal Society A* 374:20150292.

Nicol, S., A. Bowie, S. Jarman, D. Lannuzel, K. M. Meiners, and P. van der Merwe. 2010. Southern Ocean iron fertilization by baleen whales and Antarctic Krill. *Fish and Fisheries*:1–7.

Pearson, H. C., M. S. Savoca, D. P. Costa, M. W. Lomas, R. Molina, A. J. Pershing, C. R. Smith, J. C. Villaseñor-Derbez, S. R. Wing, and J. Roman. 2023. Whales in the carbon cycle: Can recovery remove carbon dioxide? *Trends in Ecology & Evolution* 38:238–249.

Ratnarajah, L., A. R. Bowie, D. Lannuzel, K. M. Meiners, and S. Nicol. 2014. The biogeochemical role of baleen whales and krill in Southern Ocean nutrient cycling. *PLoS One* 9:e114067.

Ryan, C., M. Santangelo, B. Stephenson, T. A. Branch, E. A. Wilson, and M. S. Savoca. 2023. Commercial krill fishing within a foraging supergroup of fin whales in the Southern Ocean. *Ecology* 104:e4002.

Savoca, M. S., M. F. Czapanskiy, S. R. Kahane-Rapport, W. T. Gough, J. A. Fahlbusch, K. C. Bierlich, P. S. Segre, J. Di Clemente, G. S. Penry, D. N. Wiley, J. Calambokidis, D. P. Nowacek, D. W. Johnston, N. D. Pyenson, A. S. Friedlaender, E. L. Hazen, and J. A. Goldbogen. 2021. Baleen whale prey consumption based on high-resolution foraging measurements. *Nature* 599:85–90.

Reviewer #2 (Remarks to the Author):

The Gilbert et al. manuscript is a very interesting and important modelling study of the likely sources of variability in animal-mediated nutrient cycling among a large range of ocean systems. The results of the models provide a formal assessment and presentation of hypotheses about the relative importance of small, deep diving and surface feeding cetaceans among systems. Sensitivity analysis of the modelled results point to two very important sources of error in the estimates, abundance estimates of the whales and dolphins and nutrient concentrations of the prey field, as well as consequences for prey switching. The modelling methodology is sound, although I think it will be very important to expand on the unknowns in the model framework, assumptions (i.e. fixed diet) and large sources of variability highlighted in the models. A section on unknowns and refinements in the model would be well placed in the Discussion.

Minor points (eidts)

L13 "eight nutrient cycling" word misplaced?

L32-33 the jump between "carbon capture" and planet climate should be better explained with examples of storage and sequestration linked to nutrient cycling

L325 "to to" repeat

Table 2 Missing estimate for N in Mediterranean from this study?

Fig 2 M² should be in superscript

Fig 5 is very hard to read, should be cleaned up or simplified

Reviewer #3 (Remarks to the Author):

General comments:

This paper sought to evaluate how cetaceans' contribution to nutrient cycling (two macronutrients and six micronutrients) varies geographically, and what the contribution of community composition is to this variability. This was accomplished with relatively limited existing data, which is discussed in the methods, and a modified bioenergetics model. This is the first attempt at a global study of cetaceans' role in biogeochemical cycling. I recommend this paper for publication with minor revision.

The contributions of different cetacean groups to the input of different nutrients, based on prey type, and the implications related to foraging behavior are, to me, the most noteworthy results of this work. The regional differences in nutrient inputs is less noteworthy, as it confirms known patterns of cetacean abundance in coastal zones (even the "oceanic" regions considered a not open ocean), mirroring patterns of productivity. For example, highlighting this point, the authors find that the Mediterranean had the lowest cetacean nutrient input, compared to other locations at similar latitudes (Lines 115-118). However, those similar locations have either upwelling regimes, or high-amplitude seasonal phytoplankton blooms. Further, the Mediterranean likely has depleted cetacean abundance due to a long history of whaling and relative easy access. Given that the authors found that the model is most sensitive to cetacean abundance, discussion of the factors driving abundance (e.g. whaling and other human activity like fishing, shipping), in addition to productivity (which is already included), would be good. I do think that, despite lower latitudes being more oligotrophic, cetaceans still having less of an impact on these nutrient concentrations (compared to background) than at higher latitudes is really surprising to me.

One issue that stuck with me in this paper is the over-simplistic treatment of primary production in relation to temperature and geographic region. In the introduction there is no discussion of water column structure and stratification, which is driving the temperature relationship that the authors highlight with nutrient inputs. A higher average annual temperature is generally associated with oligotrophic conditions due to near-year round stratification, limiting physical mixing and, therefore, nutrient input, thereby limiting primary production. Whales are documented in greatest abundances in regions of higher primary production because they must target energy-rich habitats to support their biomass and life histories. There is limited discussion of seasonality and this chain of causality. Treating the linear correlation between nutrient input from cetaceans and temperature as a separate result from that with primary production is misleading, since the two factors are tied together on a global scale. This needs to be addressed in the introduction, results and discussion. It might be interesting to normalize the nutrient contributions of the different whale groups by their abundance to see new group-specific patterns related to metabolism, geography, species composition in different regions, etc. A final point that is mentioned but not fully treated is that the baleen whales and delphinids are cycling nutrients within the euphotic zone. Deep divers, on the other hand, are introducing nutrients from the deep – having potentially a disproportionate impact on nutrients in the EZ. On top of this, these deep divers are

hard to quantify given their elusive lifestyle. I think this is worth mentioning as a caveat of abundance data.

The methods are well-documented and the methodology is sound. The authors did a good job justifying methodological decisions considering the limited data with which they were working. Given the extent to which data are extrapolated (with solid reasoning) in order to meet the needs of the model, some discussion of what further studies ought to be undertaken to make the estimates in this paper more accurate would be nice. I don't think this extrapolation would prohibit publication, but it needs to be addressed in the discussion, as well as the methods (where it already is). Some copy-editing is needed.

Specific comments:

Introduction

Lines 63-73: This text seems a bit over-simplified, and vague. It would be better if some numbers or percentages of global primary production from different regions could be included.

Results

Figure 1 is fantastic! In Figure 2, it's pretty clear that chl and temp are correlated, as would be predicted on average. The slopes and relative intercepts of the lines are remarkably mirrored, because temperature and primary production are linked at a global-annual scale. In Figure 3, what does the green shading indicate?

Discussion

The discussion of relative contribution of the 3 types of cetaceans would benefit from some normalization to whale type biomass; in other words, are these differences solely attributable to the abundance of the different whale types in these areas? Or something that scales non-linearly with biomass, like metabolic rate, for example?

Given the extent to which data are extrapolated in order to meet the needs of the model, some discussion of what further studies ought to be undertaken to make the estimates in this paper more accurate would be nice.

Point-by-point response to reviewers

1. General response to reviewers and overview of the major revisions

We would like to thank the three reviewers for their careful reading and constructive comments. They were very helpful to significantly improve our manuscript **Composition of cetacean communities worldwide shapes their contribution to ocean nutrient fertilisation**. We believe the manuscript is now clearer in reporting our findings and placing them in the context of nutrient cycling in the worlds' marine ecosystems.

Revisions included:

- Introduction: this section has been significantly amended. Existing material was restructured and reworded. Also, some material has been added to (i) introduce the characteristics of cetaceans that make them unique nutrient vectors, (ii) explain that their contribution to nutrient cycling is likely to be small at large scales when compared to some other processes, but that it could be important at local scales, and (iii) explain that the nutrients released by cetaceans that are not directly taken up by primary producers could affect productivity through other pathways, so that quantifying the amount of nutrients they release is a good first approach to getting a sense of their role in nutrient cycling.
- Results: minor revisions have been conducted in this section following comments from the reviewers.
- Discussion: this section has been significantly amended, following comments from the reviewers. It was restructured and reworded based on the existing material, but some new material was added to expand on (i) how the different phenology of the small delphinids, deep divers and baleen whales may shape the temporal variability in the amounts of nutrients they release (ii) how patterns of habitat use and social aggregation/segregation behaviour may shape the patterns of cetacean nutrient deposition in the environment, and (iii) what studies may be undertaken to go further considering the bias and limits in our approach.
- Methods: only minor revisions have been conducted in this section following comments from the reviewers.
- Tables and figures: Fig. 4 has been significantly modified to facilitate reading and interpretation, following the comment of reviewer #2. We added a number from our study that was missing in Table 2, as pointed out by reviewer #2. Still in Table 2, we added figures from Roman and McCarthy (2010) that were missing (atmospheric N deposition), and we specified the surfaces of the areas of the Northwest Atlantic in our study and in Roman McCarthy, to facilitate comparison. Fig.2, 3 and 7 were only slightly modified (typeface and size, superscripts, variables names) to improve readability.
- Citations and reference list: we re-set citations automatically throughout the manuscript using Nature journal style.
- Language: We have revised our wording throughout the manuscript to clarify, simplify and improve readability.

Comments and concerns of reviewers were addressed in detail below. Throughout the rest of the document, references to the manuscript are to the revised version, i.e. the version with tracked changes but with Simple Markup. Our responses are in blue.

2. Point-by-point response to reviewer #1

Reviewer #1 (Remarks to the Author):

This paper takes on a broad, difficult, and important task of attempting to quantify the role of cetaceans in global ocean nutrient cycling. The authors take on this ambitious task across all clades of cetaceans and in all ocean basins. There are a lot of strong ideas and analyses in the present draft, but what is lost with an analysis at this level is the granular detail that may 'make or break' these findings. Below I discuss several major concerns with the present paper. After that I touch on minor concerns that are easily remedied.

One major concern I have is the spatial scales employed in this study. Research has shown that when viewed at the scale of oceans, it is hard to make top predator nutrient recycling "matter" when compared to abiotic sources of nutrient addition and mixing (Maldonado et al. 2016). Instead, cetaceans and other top predators, may initiate or enhance spatially restricted hot spots or "hot moments" (Pearson et al. 2023). I think adding that level of granularity to this manuscript would be immensely helpful. For example, large groups of baleen whales in the modern Southern Ocean can number several hundred to 1000 individuals (Herr et al. 2022, Ryan et al. 2023); how much trace metals might they recycle on the scale of days and tens of km? How do those levels compare to background trace metal concentrations in that same region at that same time of year without whales present?

We agree that the spatial (and annual) scale associated to our estimates is unlikely to fit with the very local (and transient) scale of a defecation event by cetaceans. However, we did not necessarily mean to make cetacean nutrient recycling matter. Rather, we intended to broaden the picture of their contribution to nutrient cycling, starting at the scale of large ocean basins. When cetacean fertilisation is minor over large spatial scales compared to that mediated by bacteria and plankton (Maldonado et al. 2016, now cited in the manuscript as reference 23) or abiotic processes (e.g. Smith et al. 2021, reference 22 in the manuscript), we agree that cetacean fertilisation could still matter on smaller spatial and temporal scales, and we agree that it was not clearly stated and emphasized in the first version of the manuscript. With the major revision of the manuscript, we think it is now one of the key take-home messages. We have included several explicit statements on these aspects.

In the introduction:

- while mentioning the characteristics of cetaceans that make them unique nutrient vectors, we mention their tendency to form aggregations, and possibly hotspots and hot moments of nutrient cycling (L63-64)
- We explicitly state that the contribution of cetaceans is probably minor at large scales when compared to the biological cycling mediated by the microfauna and the nutrient supply provided by physical processes – but that it could be important on local scales L66-71

In the discussion:

- we say that the ecological importance of cetacean fertilisation is as relative as their contribution to shape the nutrient background in their environment, and thus relative to other nutrient inputs L272-273 (and throughout the following paragraph)
- the paragraph L380-400 discusses patterns of aggregation and dispersion and potential implications in terms of nutrient transfers and recycling

At the scale the data is currently presented (vast ocean regions) I am left thinking the correlation of whale nutrient release and remotely-sensed chlorophyll is likely more bottom up than top down. Without examining this relationship in more detail, both spatially and temporally, it is hard to assign

any importance to cetaceans. Meaning, at large spatial scales in an eastern boundary upwelling system, for example, upwelling itself drives the majority of primary productivity, and whales (and other predators) show up 1-2 months after peak phytoplankton blooms to forage on low trophic level prey that was fed by the upwelling-fueled bloom (Croll et al. 2005).

We did not mean that the correlation between levels of nutrient release and productivity reflected a top-down process, we actually meant the opposite (*"As predators depend on lower trophic levels, cetacean populations reflect the state of their environment"*, now L237-238 but L226-227 in the first version of the manuscript). We understand that the mention of "virtuous cycle" may have been the source of the confusion, which was also pointed out by reviewer #3. We did not mean the cycle went up to the uptake of nutrients released by cetaceans by primary producers (*"However, the quantities of nutrients released back in ecosystems by cetaceans might not directly reflect primary producers uptake as well as cetaceans' relative contribution to ecosystem productivity"* in the first version L235-237, now in the paragraph L248-251, but reworded). However, we recognise that our wording was ambiguous in some parts.

We have modified this section on the link between levels of nutrients released by cetaceans and indicators of productivity. It is now explicit from the beginning of the discussion that we interpret this correlation as a bottom-up relation and not a top down one L235-238. We reformulated the introduction of the nutrient virtuous cycle to better link it with our results L238-240, and we explicitly say that this this bottom-up process does not necessary scale with the top-down effect cetacean fertilisation may have on productivity L248-251.

Additionally, ecosystem context is not provided in the level of detail it needs to be; in several cases this leads you to compare apples to oranges. For example, in lines 97-114, you compare nutrient release rates between micronutrients (e.g., trace metals) and macronutrients (N and P). These are not equivalent. One gram of bioavailable iron addition to iron-limited waters can stimulate several kilograms of primary production, the same is not true of N or P in waters limited by those macronutrients. If you plan to make these comparisons, it needs to be done in relation to the amount of primary production the levels of whale-recycled nutrients can seed.

We agree that it was not appropriate to compare scales of release of N and Co in the Results. This has been modified L121-125.

Similarly, all these comparisons need to be done relative to the nutrient that is limiting in the ecosystem. This needs to be done on an ecosystem-by-ecosystem basis. For example, the additional N and P that whales recycle in the Southern Ocean is effectively meaningless in HNLC waters, whereas the amount of trace metals, specifically iron, they recycle (in a trace metal-limited system) could be immensely important. This is why there has been so much work done on the latter (Nicol et al. 2010, Lavery et al. 2010, 2014, Ratnarajah et al. 2014, Savoca et al. 2021), but not the former.

We agree this important functional point was not clearly addressed in the first version of the manuscript. In the discussion L251-252, we now emphasize that depending on local demand, not all nutrients released by cetaceans necessarily scale with ecosystem response, and we develop this aspect with examples L252-258. The ecosystem context is also discussed in the following paragraph L272-294 on how cetaceans may participate in shaping the nutrient background in their environment. We think it is now clearer that local conditions vary between ecosystem and habitats and that it needs to be taken into account when assessing the role and importance of cetacean fertilisation.

Finally, some relevant cetacean biology and ecology is lacking. The differences in annual phenology of cetacean feeding is glossed over, but is immensely important for this paper. Baleen whales have a

compressed but very intense feeding season at high latitudes. Toothed whales generally eat similar amounts throughout the year, and across the world's oceans at all latitudes. Most baleen whales migrate; many toothed whales do not have the same predictable, long migrations. Point being, if you're looking at the total amount of nutrient(s) recycled on an annual basis and conclude that it is similar for baleen and toothed whales, but do not explicitly consider that baleen whales do nearly all their feeding in a 4-month period (vs spread out evenly across the entire year for most toothed whales), this may lead to erroneous conclusions about the "value" of that potential fertilizer.

While we had this in the back of our mind, we agree that we failed to make it explicit in the first version of the manuscript. The phenology is now clearly discussed in the paragraph L364-379. We have also added some further development on other aspects of cetacean biology and ecology throughout the discussion (metabolism, diet, foraging ecology, social behaviour).

Minor issues:

Line 13: "eight nutrient cycling" doesn't make sense as written.

This has been corrected L13-14.

Lines 30-34 & 318-321: you mention whales as a natural climate solution (NCS) while only mentioning carbon capture and storage. NCS only happens when carbon is sequestered for hundreds or thousands of years. This is an especially important component of this whole discussion and one that is unknown for whales. This section needs to be written much more speculatively.

We have reworded the section in the introduction to better explain how primary producers are involved in the biological carbon pump L50-56, to highlight that this is one of the reasons why there is much research to identify the processes that regulate its productivity in the world's oceans. We have removed the section identifying cetaceans as NCS in the discussion, as we agree that this is highly speculative at this stage. Still, while emphasising this we say it is difficult to imagine the ecosystem service provided by cetaceans being replaced by artificial fertilisation L411-414.

Lines 62 and 64: "at large scale" and "at small scales" should likely be rephrased to: "at large/small spatial scales"

This has been modified L41 and L46-47.

Lines 182-185: Oddly worded, suggest rephrasing

We modified this sentence L208-212, and the following part to describe results with more precision.

Lines 205: But N and P are macronutrients.

This sentence has been deleted.

Lines 287-293: This is an essential point and needs to be mentioned much earlier in the paper (the introduction) and reiterated here.

The whale pump is now mentioned in the introduction L61-63. However, we did not mention the turnover rate in the introduction. While the turnover is relatively short when predators feed on zooplankton, it is still longer than the turnover of nutrients recycled by the microbial community or zooplankton itself, which is one of the reasons that make the contribution of predators minor over large scales (Maldonado et al. 2016). Still, it is a characteristic that differ between cetacean taxa, so we think it is well placed in the discussion – and it is discussed now L330-341.

Line 325: delete one "to"

The conclusion has been modified L401-422, and this sentence has been deleted.

Line 409: provide scientific names when first introducing species.

This has been corrected L491-492.

3. Point-by-point response to reviewer #2

Reviewer #2 (Remarks to the Author):

The Gilbert et al. manuscript is a very interesting and important modelling study of the likely sources of variability in animal-mediated nutrient cycling among a large range of ocean systems. The results of the models provide a formal assessment and presentation of hypotheses about the relative importance of small, deep diving and surface feeding cetaceans among systems. Sensitivity analysis of the modelled results point to two very important sources of error in the estimates, abundance estimates of the whales and dolphins and nutrient concentrations of the prey field, as well as consequences for prey switching. The modelling methodology is sound, although I think it will be very important to expand on the unknowns in the model framework, assumptions (i.e. fixed diet) and large sources of variability highlighted in the models. A section on unknowns and refinements in the model would be well placed in the Discussion.

The higher sensitivity of the model to the diet means nutrient concentration for trace nutrients than for major nutrients is discussed L320-322. Following the recommendation of reviewer #2 (and #3), we emphasised how variability may have been underestimated in our approach to clearly identify what parameters should primarily be more finely set for studies at more local scales L322-329. We modified the section explaining how we set a fixed diet in the Methods to highlight how we limited the extent of this bias L501-507. We also explain L356-363 that the biochemical properties of the released wastes are not included in our approach, but may be particularly important in determining the fate of the released nutrients in ecosystems L356-363. Refining parameters such as population abundance is indeed an important research goal in the context of population assessments, and most of the sources of unknown associated to these estimates are due to field parameters that are hardly controllable (animal occurrence and surface availability, bad conditions, etc). Similarly, if it would be important to get estimates of nutrient release rates, it would be especially challenging to obtain for cetaceans in the wild– if it is even feasible. Furthermore, we also think further studies at more local scales (as said now L326-329), or studies exploring the biochemical properties of cetacean wastes (L360-363) would be valuable, and at such spatial scale more precise abundance and nutrient data are mandatory.

Minor points (edits)

L13 "eight nutrient cycling" word misplaced?

This has been corrected L13-14.

L32-33 the jump between "carbon capture" and planet climate should be better explained with examples of storage and sequestration linked to nutrient cycling

We have reworded this section to better explain how primary producers are involved in the biological carbon pump, to highlight why there is so much research to identify the processes that regulate its productivity in the world's oceans L50-56.

L325 "to to" repeat

The conclusion has been modified L401-422, and this sentence has been deleted.

Table 2 Missing estimate for N in Mediterranean from this study?

Indeed, we thank reviewer #2 for pointing this out! This has been corrected Table 2, L754.

Fig 2 M^2 should be in superscript

This has been corrected on Figure 2, L767 (plus mg/m^3 for chlorophyll concentration).

Fig 5 is very hard to read, should be cleaned up or simplified

Figure 5 is now Figure 4 (as it was mentioned first in the Results). We agree this figure was hard to read. We have modified it by removing the name of variables on arrows (representing variables, i.e. statistics of estimates per nutrient), but colouring the arrows instead so that each is associated to a nutrient. We think it is now better (Figure 4, L800).

4. Point-by-point response to reviewer #3

Reviewer #3 (Remarks to the Author):

General comments:

This paper sought to evaluate how cetaceans' contribution to nutrient cycling (two macronutrients and six micronutrients) varies geographically, and what the contribution of community composition is to this variability. This was accomplished with relatively limited existing data, which is discussed in the methods, and a modified bioenergetics model. This is the first attempt at a global study of cetaceans' role in biogeochemical cycling. I recommend this paper for publication with minor revision.

The contributions of different cetacean groups to the input of different nutrients, based on prey type, and the implications related to foraging behavior are, to me, the most noteworthy results of this work. The regional differences in nutrient inputs is less noteworthy, as it confirms known patterns of cetacean abundance in coastal zones (even the "oceanic" regions considered a not open ocean), mirroring patterns of productivity. For example, highlighting this point, the authors find that the Mediterranean had the lowest cetacean nutrient input, compared to other locations at similar latitudes (Lines 115-118). However, those similar locations have either upwelling regimes, or high-amplitude seasonal phytoplankton blooms. Further, the Mediterranean likely has depleted cetacean abundance due to a long history of whaling and relative easy access. Given that the authors found that the model is most sensitive to cetacean abundance, discussion of the factors driving abundance (e.g. whaling and other human activity like fishing, shipping), in addition to productivity (which is already included), would be good.

We agree with the arguments raised by the reviewer, however, we feel that a detailed area-per-area analysis would lengthen and complicate the discussion. Treating homogeneously the spatial differences in levels of nutrient release based on processes conditioning productivity, together with factors that may drive (or may have driven) current cetacean abundances would be challenging and require data which are not systematically available for each area. For example, discussing why levels are lower in the Mediterranean Sea than at similar latitudes would also require discussing why levels are higher in the Southwest Indian Ocean than at similar latitudes. Similarly, whaling histories are different depending on locations. There also has been a long and intense whaling activity on the shores of the Atlantic Ocean, which likely impacted current abundances of hunted species in the Northeast,

Northwest and central North oceans areas. Some species have been severely depleted, or even whaled to extinction (e.g. North Atlantic right whale used to be present in the Northeast Atlantic Ocean, hunted intensely by the Basques). Some populations recovered (e.g. humpback whales), some other still struggle (e.g. North Atlantic Right whales again, now limited to the Northwest Atlantic region). Fin and minke whales are still hunted to some extent in the central North Atlantic, and delphinids species such as long-finned pilot whales are still taken occasionally in the Faroes islands in the traditional “grindadráp”. We feel it could be treated in an entirely dedicated analysis. Our focus is more on assessing the role of cetacean populations today, at their current abundance in the world’s oceans.

We adopted a more general approach to better emphasise aspects that should be accounted for a more local examination. It is now explicitly said that levels of nutrient release by cetacean mirrors patterns of productivity L235-238 as the first point in the discussion (and in the Abstract L15-16). It is also clearly emphasized that the ecosystem context of the recipient ecosystem (local demand L248-271 and local nutrient background L272-294) needs to be taken into account in the assessment of the role of cetaceans in nutrient cycling.

I do think that, despite lower latitudes being more oligotrophic, cetaceans still having less of an impact on these nutrient concentrations (compared to background) than at higher latitudes is really surprising to me.

We did not mean that the impact of cetacean fertilisation was lower in oligotrophic regions than in meso-to-eutrophic regions in higher latitudes – we meant quite the opposite, in fact (see *“However, the quantities of nutrients released back in ecosystems by cetaceans might not directly reflect primary producers up-take as well as cetaceans’ relative contribution to ecosystem productivity”* L235-237 in the first version of the manuscript, now reworded L237-238, and *“It is likely that primary producers from oligotrophic ecosystems rely more on animal-mediated nutrient cycling than in meso- to eutrophic ecosystems where nutrients are largely supplied by other sources”* L240-245 in the first version, now reworded L274-276). However, we understand this must have been ambiguous in the first version of the manuscript due to a possible misinterpretation of what we meant when mentioning a “nutrient virtuous cycle”. We think it is now clearer that we interpret the correlation between levels of nutrient release by cetaceans and indicators of productivity as a bottom-up process and not a top-down one, as it is explicitly said L235-238. We also think it is clearer that the contribution of cetaceans to stimulating primary producers’ productivity is likely greater in oligotrophic areas than in areas already repleted in nutrients (see L276-280, L353-355), even if this cannot be quantified nor deduced directly from our results.

One issue that stuck with me in this paper is the over-simplistic treatment of primary production in relation to temperature and geographic region. In the introduction there is no discussion of water column structure and stratification, which is driving the temperature relationship that the authors highlight with nutrient inputs. A higher average annual temperature is generally associate with oligotrophic conditions due to near-year round stratification, limiting physical mixing and, therefore, nutrient input, thereby limiting primary production. Whales are documented in greatest abundances in regions of higher primary production because they must target energy-rich habitats to support their biomass and life histories. There is limited discussion of seasonality and this chain of causality. Treating the linear correlation between nutrient input from cetaceans and temperature as a separate result from that with primary production is misleading, since the two factors are tied together on a global scale. This needs to be addressed in the introduction, results and discussion.

We did not mean to treat separately the correlation between levels of cetacean fertilisation and sea surface temperature and the correlation between levels of cetacean fertilisation and mean surface

chlorophyll concentration. We chose these two parameters because they are both indicators of ecosystem productivity, and because they are often included in habitat models. We chose to include both and not only one as we aimed to test how strong the correlations were with these two indicators, one being a direct indicator (chloro) and the other one more of determinant factor (sst). We recognise that they are strongly correlated (negatively), as pointed out by reviewer #3, and that we failed to explicitly say this in the first version of the manuscript. This has been added in the Results L139-141. In the introduction, we added that tropical systems are characterised by an intense stratification of the water column L42-44 and that there are some intense seasonal variations in productivity levels in temperate regions L44-46. In the discussion, we now mention the stratification of the water column as an important factor in the inherent characteristic of the recipient ecosystem L276-278 and L353-355. The seasonality is discussed in the paragraph examining differences in the phenology of the three taxa L364-379.

It might be interesting to normalize the nutrient contributions of the different whale groups by their abundance to see new group-specific patterns related to metabolism, geography, species composition in different regions, etc.

We followed reviewer #3 recommendation and made a couple of trials. We made biplots with the mean relative contribution to total nutrient release in each area and the relative abundance and relative biomass of each taxon (see figure below). Indeed, we see some group-specific patterns, displaying expected patterns:

- Relative contribution/relative abundance: slope is strongest for baleen whales, as for a same number of individuals, whales obviously release more nutrients than dolphins due to the large differences in their body mass and thus levels of food consumption.
- Relative contribution/relative biomass: slope is equivalent for all three taxa, but the y-intercept is highest for small delphinids and lowest for baleen whales, i.e. for an equivalent biomass small delphinids release more nutrients than baleen whales. This is due to differences in their metabolism (quantified in the parameter Beta in our model). Baleen whales consume more food daily than small delphinids, individually, but they actually have a lower consumption rate per unit of body mass due to their lower metabolism.

We now discuss these aspects L299-305, and these differences in the levels of consumption and nutrient release are accessible per species in supplementary material 3 (% of body mass consumed daily, and amounts of nutrient ingested and released daily per individual).

A final point that is mentioned but not fully treated is that the baleen whales and delphinids are cycling nutrients within the euphotic zone. Deep divers, on the other hand, are introducing nutrients from the deep – having potentially a disproportionate impact on nutrients in the EZ. On top of this, these deep divers are hard to quantify given their elusive lifestyle. I think this is worth mentioning as a caveat of abundance data.

We think the vertical transfer of nutrient that can be mediated by cetaceans is now more clearly emphasised, with the mention of the “whale pump” in the introduction L61-63 and with the paragraph dedicated to this aspect in the discussion L342-355. However, we did not go into the details of the factors that could influence the abundance estimates of the three taxa. The elusive lifestyle of deep divers is usually included in these estimates through availability correction factors, which often results in high coefficient of variations. We used these coefficients of variation for Monte Carlo simulations in our model, so this is directly reflected in our results with the large confidence intervals for the contribution of deep divers in most areas (supplementary material 1).

The methods are well-documented and the methodology is sound. The authors did a good job justifying methodological decisions considering the limited data with which they were working. Given the extent to which data are extrapolated (with solid reasoning) in order to meet the needs of the model, some discussion of what further studies ought to be undertaken to make the estimates in this paper more accurate would be nice. I don't think this extrapolation would prohibit publication, but it needs to be addressed in the discussion, as well as the methods (where it already is).

This comment is addressed in the response to reviewer #2.

Some copy-editing is needed.

We have revised the language throughout the manuscript.

Specific comments:

Introduction

Lines 63-73: This text seems a bit over-simplified, and vague. It would be better if some numbers or percentages of global primary production from different regions could be included.

We have modified this section L39-50. We did not include numbers, but we now provide ratios to compare the productivity in tropical and temperate regions, taken from the maps in Longhurst et al 1995 (reference 34 in the manuscript).

Results

Figure 1 is fantastic! In Figure 2, it's pretty clear that chl and temp are correlated, as would be predicted on average. The slopes and relative intercepts of the lines are remarkably mirrored, because temperature and primary production are linked at a global-annual scale. In Figure 3, what does the green shading indicate?

We appreciate the enthusiasm of the reviewer!

For Figure 2, as said above, it is now clearly said in the Results L139-141 that the two indicators are correlated.

Green shading on Figure 3 are violin plots, they display the distribution of values – we indeed failed to mention this in the caption. This has been added L787.

Discussion

The discussion of relative contribution of the 3 types of cetaceans would benefit from some normalization to whale type biomass; in other words, are these differences solely attributable to the abundance of the different whale types in these areas? Or something that scales non-linearly with biomass, like metabolic rate, for example?

See above for our response to this interesting comment.

Given the extent to which data are extrapolated in order to meet the needs of the model, some discussion of what further studies ought to be undertaken to make the estimates in this paper more accurate would be nice.

We also responded to this comment above.

REVIEWERS' COMMENTS

Reviewer #1 (Remarks to the Author):

I appreciate the author's careful consideration of the points I raised. It appears the line numbers in the pdf I received and the line numbers in the response to reviewers do not correspond (due to a continuous line numbering error from page to page). I have encountered this in my own papers; it is frustrating and not the authors fault, though it does make scrutinizing what portions they changed due to my feedback difficult.

The introduction is reworked in that it places the study in better context, which I deemed as an essential change from the first version of the paper I reviewed. It seems the authors have done that.

I do appreciate the reworked Results section; it is well organized with the subheadings.

Overall, my main remaining critique boils down to the title suggesting that the paper will report how cetaceans contribute to ocean nutrient "fertilization," but the data to perform or model those effects are limited, and none are applied here.

For examples of how they may be applied, see Ratnarajah et al 2016 Ecol modeling, Lavery et al. 2010 Proc B, Roman et al. 2016 PLoS One, or Savoca et al. 2021 Nature (extended data table 2 specifically). While I do think this approach of how nutrients may actually "fertilise" marine primary producers would be an exciting application, particularly across regions and species as is attempted in the present paper, it would require extra work that is likely outside the scope of the present manuscript.

Therefore, the simplest way to deal with this is replace "fertilisation" with "cycling" in the title. A title of: "Composition of cetacean communities worldwide shapes their contribution to ocean nutrient cycling," would more accurately reflect what the study actually accomplishes.

With that change, I would be more comfortable suggesting this paper be accepted. I am happy to review a later revision of this work, but it seems like a title change as suggested to accompany the already-made changes to this version would be sufficient.

Reviewer #2 (Remarks to the Author):

Thank you for the careful revision of your manuscript 'Composition of cetacean communities worldwide shapes their contribution to ocean nutrient fertilisation'. I was pleased to see careful consideration and modifications made to each of the review comments. The methods, analyses, and reporting are all sound. The work is timely and noteworthy in that the role of large animals in functional connectivity and as vectors of limiting nutrients is being increasingly appreciated in the literature. The argument is always that perhaps the contribution of an individual taxonomic group (whales) to nutrient recycling is small compared to abiotic factors, but I think when you combine whales into a category of biological nutrient vectors along with large and even small fishes, sea birds, other marine mammals that the collective process is indeed large. The current manuscript provides just that direction of inquiry by considering toothed whales and their contribution to the process. While many of these animals are relatively small compared to the baleen whales they generally feed at higher trophic levels, and in a system where essential micronutrient can be bioaccumulated within food webs their contribution can be proportionally large. These are essential and important considerations in the context of conservation and management of ocean resources and the services they provide.

Reviewer #3 (Remarks to the Author):

I appreciate the care the authors took to revise their manuscript according to the reviewer comments, as well as the extra analysis that they did in response to reviewer comments. The manuscript is acceptable to me for publication.

Point-by-point response to reviewers

Again, we would like to thank the three reviewers for their careful reading and constructive comments. We addressed the point they raised and think our **Composition of cetacean communities worldwide shapes their contribution to ocean nutrient cycling** is now definitely clearer in reporting our findings and placing them in the context of nutrient cycling in the worlds' marine ecosystems.

Revisions of the manuscript in light of reviewers' comments are limited to the change of the title, following suggestion of reviewer #1. Other slight modifications were made in response to points raised by the editorial staff and are detailed in the completer author's checklist. We provided one version with tracked-changes 'Nutricet_revision_3rd-round_tracked_changes.docx' and one version with all tracked-changes accepted 'Nutricet_revision_3rd-round.docx'.

Comments and concerns of reviewers were addressed in detail below. Our responses are in blue.

1. Point-by-point response to reviewer #1

Reviewer #1 (Remarks to the Author):

appreciate the author's careful consideration of the points I raised. It appears the line numbers in the pdf I received and the line numbers in the response to reviewers do not correspond (due to a continuous line numbering error from page to page). I have encountered this in my own papers; it is frustrating and not the authors fault, though it does make scrutinizing what portions they changed due to my feedback was difficult.

The introduction is reworked in that it places the study in better context, which I deemed as an essential change from the first version of the paper I reviewed. It seems the authors have done that.

I do appreciate the reworked Results section; it is well organized with the subheadings. Overall, my main remaining critique boils down to the title suggesting that the paper will report how cetaceans contribute to ocean nutrient "fertilization," but the data to perform or model those effects are limited, and none are applied here.

For examples of how they may be applied, see Ratnarajah et al 2016 Ecol modeling, Lavery et al. 2010 Proc B, Roman et al. 2016 PLoS One, or Savoca et al. 2021 Nature (extended data table 2 specifically). While I do think this approach of how nutrients may actually "fertilise" marine primary producers would be an exciting application, particularly across regions and species as is attempted in the present paper, it would require extra work that is likely outside the scope of the present manuscript.

Therefore, the simplest way to deal with this is replace "fertilisation" with "cycling" in the title. A title of: "Composition of cetacean communities worldwide shapes their contribution to ocean nutrient cycling," would more accurately reflect what the study actually accomplishes.

With that change, I would be more comfortable suggesting this paper be accepted. I am happy to review a later revision of this work, but it seems like a title change as suggested to accompany the already-made changes to this version would be sufficient.

We appreciated reviewer #1 feedback, and we were sorry to read that the review has been complicated by line numbering issues.

We agree with the points raised about the use of 'fertilisation' in the title, it suggested we went a step further than we actually did. We made the change.

2. Point-by-point response to reviewer #2

Reviewer #2 (Remarks to the Author):

Thank you for the careful revision of your manuscript 'Composition of cetacean communities worldwide shapes their contribution to ocean nutrient fertilisation'. I was please to see careful consideration and modifications made to each of the review comments. The methods, analyses, and reporting are all sound. The work is timely and noteworthy in that the role of large animals in functional connectivity and as vectors of limiting nutrients is being increasingly appreciated in the literature. The argument is always that perhaps the contribution of an individual taxonomic group (whales) to nutrient recycling is small compared to abiotic factors, but I think when you combine whales into a category of biological nutrient vectors along with large and even small fishes, sea birds, other marine mammals that the collective process is indeed large. The current manuscript provides just that direction of inquiry by considering toothed whales and their contribution to the process. While many of these animals are relatively small compared to the baleen whales they generally feed at higher trophic levels, and in a system where essential micronutrient can be bioaccumulated within food webs their contribution can be proportionally large. These are essential and important considerations in the context of conservation and management of ocean resources and the services they provide.

We appreciated reviewer #2 feedback and for the insightful comments on our work and its significance.

3. Point-by-point response to reviewer #3

Reviewer #3 (Remarks to the Author):

I appreciate the care the authors took to revise their manuscript according to the reviewer comments, as well as the extra analysis that they did in response to reviewer comments. The manuscript is acceptable to me for publication.